# Preeclampsia promotes autism in offspring via maternal inflammation and fetal NFκB signaling

Xueyuan Liu[1,2,*] , Haiyan Liu[1,*], Nihao Gu[3] , Jiangnan Pei[1], Xianhua Lin[1] , Wenlong Zhao[2,3] 

**Preeclampsia (PE) is a risk factor for autism spectrum disorder (ASD) in offspring. However, the exact mechanisms underlying the impact of PE on progeny ASD are not fully understood, which hinders the development of effective therapeutic approaches. This study shows the offspring born to a PE mouse model treated by N$^{\omega}$-nitro-L-arginine methyl ester (L-NAME) exhibit ASD-like phenotypes, including neurodevelopment deficiency and behavioral abnormalities. Transcriptomic analysis of the embryonic cortex and adult offspring hippocampus suggested the expression of ASD-related genes was dramatically changed. Furthermore, the level of inflammatory cytokines TNFα in maternal serum and nuclear factor kappa B (NFκB) signaling in the fetal cortex were elevated. Importantly, TNFα neutralization during pregnancy enabled to ameliorate ASD-like phenotypes and restore the NFκB activation level in the offspring exposed to PE. Furthermore, TNFα/NFκB signaling axis, but not L-NAME, caused deficits in neuroprogenitor cell proliferation and synaptic development. These experiments demonstrate that offspring exposed to PE phenocopies ASD signatures reported in humans and indicate therapeutic targeting of TNFα decreases the likelihood of bearing children with ASD phenotypes from PE mothers.**

## Introduction

Autism spectrum disorder (ASD) is an early-onset developmental disorder in children, characterized by deficits in socialization, restricted, repetitive behaviors, and alterations in brain size and dendritic/synaptic development (1, 2, 3, 4). The prevalence of ASD is increasing each year, with rates reaching ~1 in 44 children in the US (3, 5). Although genetics is considered to be a dominant factor in the etiology of ASD, environmental factors may contribute between 17 and 50% (6). Previous studies suggested that the uterine environments caused by preeclampsia (PE) may result in ASD in offspring (7, 8).

PE is a severe hypertensive disorder of pregnancy (HDP) that affects ~5% of pregnant women, and is a leading cause of maternal morbidity and mortality (9). Clinically, it manifests as new-onset hypertension (blood pressure ≥ 140/90 mmHg) after 20 weeks of gestation and is usually concomitant with proteinuria organ dysfunction or abnormal placentation (10). It is evident that PE is not only associated with adverse maternal outcomes including seizure, stroke, low platelet count (HELLP) syndrome, and heightened risk for maternal cardiovascular disease later, but also linked to cerebral palsy, ASD, and attention-deficit/hyperactivity disorder in exposed offspring (11, 12, 13). Mounting epidemiological evidence has shown that exposure to PE significantly increases the risk of ASD in offspring born either preterm or at term (6, 11, 14). For example, a meta-analysis systematically reviewed the association between HDP and ASD, suggesting that exposure to HDP was associated with 35% increased odds of ASD, compared with non-exposure (6). Furthermore, results from a Swedish birth cohort including 1,085,024 individuals born between 1987 and 1996 demonstrated HDP heightened the risk of ASD by 32% (hazard ratio [HR], 1.32; 95% CI, 1.23–1.42) (12). Notably, the only effective therapy for PE is induced delivery, which results in preterm birth and escalates the risk for aberrant neurodevelopment in offspring (15). In addition, prior research assumed that maternal inflammation might play a crucial role in the adverse neurodevelopment outcomes of offspring born to PE mothers. Results from a population-based study in Finland illustrated that the elevated level of maternal inflammatory biomarkers C-reactive protein in PE was remarkably linked to a 43% higher risk of autism in offspring (16, 17). Although several PE animal models have been reported to demonstrate that offspring from PE mothers indeed performed neurodevelopment and some behavioral deficiency, it still lacks further studies on PE animal models to describe the ASD-like characteristics in offspring (13, 18, 19, 20). A popular animal model of PE is established by L-N$^{G}$-nitro arginine methyl ester (L-NAME) in rodents (21, 22), which could recapitulate almost all aspects of PE pathogenesis, encompassing sustained hypertension and proteinuria (23, 24, 25). L-NAME is a molecular drug inhibiting nitric oxide synthase that supports nitric oxide yield in an endothelium system (26).

[1]Obstetrics and Gynecology Hospital of Fudan University, Shanghai, China   [2]Environmental and Occupational Health Science Institute, Rutgers University, Piscataway, NJ, USA   [3]International Peace Maternity & Child Health Hospital Affiliated to Shanghai Jiao Tong University School of Medicine and Shanghai Key Laboratory for Embryo-Feta Original Adult Disease, Shanghai Jiao Tong University, Shanghai, China

Correspondence: wz326@pharmacy.rutgers.edu; xl_1290@126.com
*Xueyuan Liu and Haiyan Liu contributed equally to this work and share first authorship

Previous research has indicated that an imbalance in pro-inflammatory and anti-inflammatory cytokines in early pregnancy is a high-risk factor for the development of ASD in offspring (27). This altered cytokine profile in maternal circulation may affect fetal brain development through indirect or direct pathways (28). It is believed that aberrant maternal immunity activation can disrupt normal fetal brain development processes such as neurogenesis and neuronal branching (29). Observation research has shown several critical pro-inflammatory cytokines such as IL-6 and TNFα are elevated in a maternal immune activation model, which has been linked to ASD (30). Several studies, including both epidemiological research and animal models, have suggested an association between TNFα and autism (31). For instance, Xiang Yu et al conducted a study where they measured plasma levels of cytokines in children with ASD and typically developing children. They found elevated levels of TNFα in male children with ASD, which positively correlated with their total development quotient (32). Another study by Jones et al demonstrated that higher levels of mid-gestational cytokines in maternal sera, including TNFα, were associated with an increased risk of ASD with intellectual disability compared to developmental delay without ASD (33). Moreover, elevated levels of TNFα have been observed in a mouse model of autism induced by prenatal exposure to valproic acid, suggesting a potential contribution of TNFα to autism in mice (34). However, no animal model has demonstrated whether common pregnancy complications such as PE or gestational diabetes increase the susceptibility of offspring to ASD through the abnormal expression of TNFα.

In this study, we focused on investigating the causal role of maternal PE in adverse neurodevelopmental outcomes of exposed offspring and the underlying mechanism. First, we established the L-NAME model of PE in C57BL/6J mice. Then, we systematically assessed whether the mouse progenies born to this model could exhibit key characterization of ASD. Furthermore, we explored the underlying mechanisms and potential therapeutic targets, thereby ameliorating autism symptoms in offspring exposed to PE.

# Results

## Establishment of a mouse model of PE

### Experimental scheme
L-NAME is a specific eNOS inhibitor commonly administered to rodents to successfully induce the critical manifestations of PE, including hypertension, kidney dysfunction, and abnormal placental development (35). Thus, we administered L-NAME (1 mg/ml) to mice on gestation day (GD) 12 through drinking water following the scheme illustrated in Fig S1A to explore the neurobiological underpinning of ASD in offspring. As expected, increased measuring systolic blood pressure (SBP) (Fig S1B) and proteinuria levels (Fig S1C) were observed in PE mother mice. There were no significant changes in water consumption between the two groups (Fig S1D).

### Intrauterine growth restriction (IUGR) and mortality in PE mouse model
We also observed that the body weights of the offspring progressively decreased from embryonic 15.5 d (E15.5) to postnatal 0 d (P0) (Fig S1E and F), and the mortality rate in the PE group at P2

was higher, compared with the control group (Fig S1G). However, the number of pups per litter, namely, the little size for each group, at P0 (Fig S1H) and gestation age were not different between the PE group and control groups (Fig S1I). These results suggested that our model mice exhibited most of the symptoms of PE and FGR without an apparent alteration in gestation age.

## Autism-like signatures in the developing brains and behavioral paradigm of offspring exposed to PE

### Inhibition of neuroprogenitor cell (NPC) proliferation in cortices of fetal mice exposed to PE
Previous studies have reported that macrocephaly and microcephaly (head circumference >97th and <3rd percentile, respectively) are observed in autistic children (4). To investigate whether neurodevelopmental alterations occur in offspring exposed to PE, we harvested brains at E15.5, E17.5, and P0. These time-points were selected because the fetal mouse brain undergoes a rapid period of neuronal proliferation, migration, and formation of synaptic connections, which have been shown to be highly relevant to autism (36, 37). Significantly, brain weight progressively decreased in the PE group (Fig 1A), and the thickness of the dorsal forebrain from the anterior-to-posterior axis significantly decreased from E17.5 onward (Figs 1B and C and S2A). In addition, the size of the striatum was also decreased at P0 (Fig S2B), and the brain/body weight ratios were increased at P0 (Fig S2C) in PE-exposed offspring. These results suggested that neurodevelopment was significantly impaired, and there was evidence of brain-sparing growth restriction in offspring exposed to PE. Next, 4-h BrdU incorporation and neurosphere culture assays were performed. Immunostaining for BrdU and phospho-histone H3 (p-H3, a mitosis marker) showed that the numbers of BrdU+ cells and p-H3+ cells were obviously decreased in the PE group from E17.5 onward (Fig 1D–F). An in vitro neurosphere assay showed that the proliferation capacity of NPC tended to be decreased at E15.5 and was significantly reduced from E17.5 (Fig 2A–C). Meanwhile, we observed that most of the cells in dissociated neurosphere were Nestin+ cells (Fig S2D and E). Hence, we postulate that the aberrant proliferation is primarily attributed to radial glial cells in offspring exposed to PE. Taken together, these results suggested that the proliferation of NPC was disrupted in PE-exposed offspring.

Next, we examined the transcriptome by RNA-sequencing (RNA-seq) analysis of E17.5 cerebral cortices to explore the underlying molecular mechanisms. The results revealed that PE exposure led to changes in the expression of 255 genes (121 up-regulated and 134 down-regulated genes) (Fig 2D and E and Table S1). Gene ontology (GO) analysis revealed that the most significantly overrepresented biological processes were the "cell cycle and stem cell maintenance," etc. (Fig 2F and Table S2). Then, six genes associated with NPC stemness maintenance or cell cycle events identified by RNA-seq, namely, *Ube2c*, *Celsr1*, *Kif18*, *Creb5*, *Gli2*, and *Plk1*, were selected for validation via Quantitative real-time PCR (qRT-PCR) and Western blot (38, 39, 40, 41, 42). The results showed that the expression of these genes in the E17.5 cortices was reduced in the PE group compared with the control group (Figs 2G and S3A). We compared the differentially expressed genes (DEGs) identified by RNA-seq with the genes in the Human Gene module of Simons Foundation

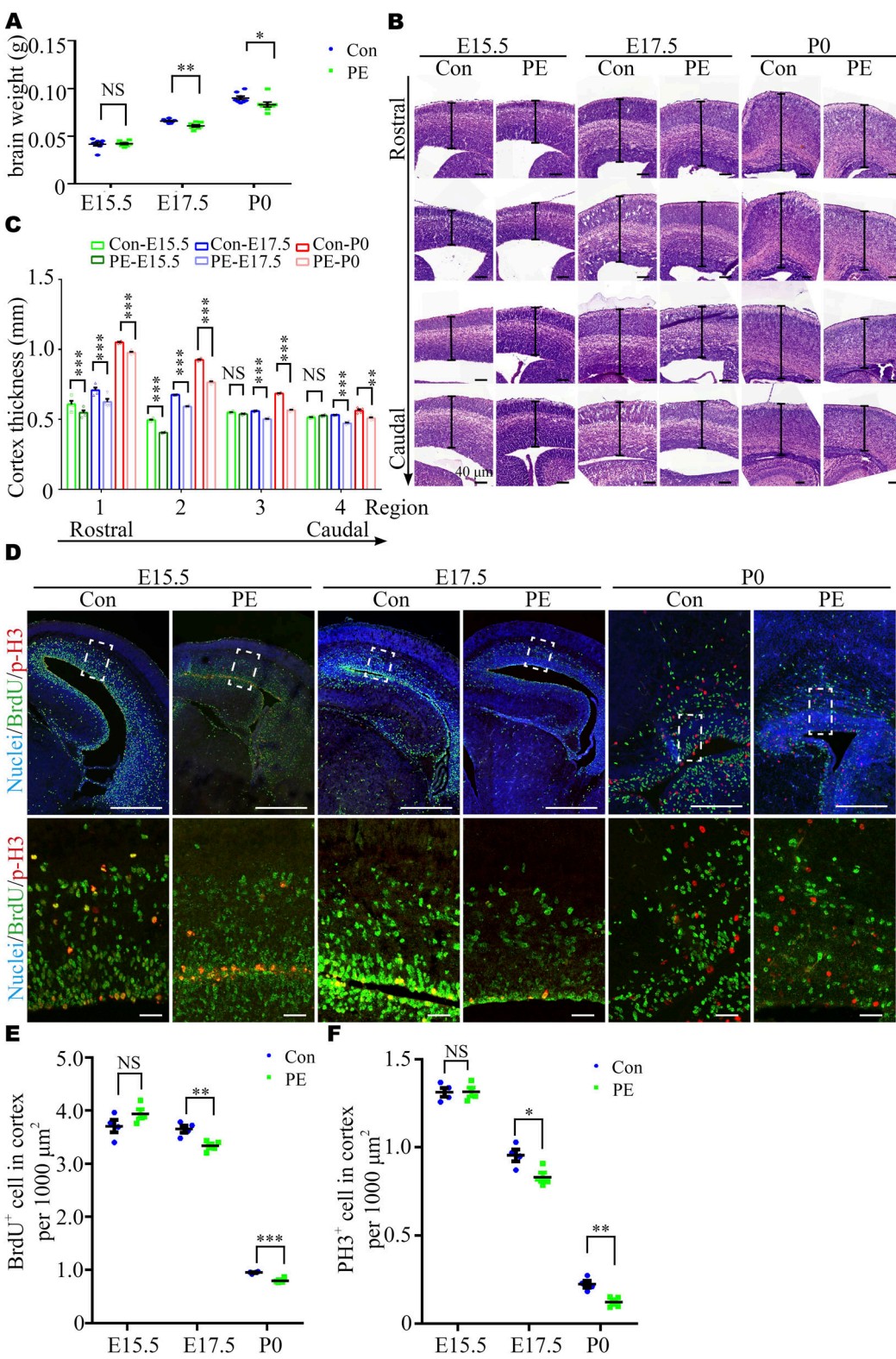

**Figure 1. Preeclampsia (PE) progressively results in aberrant brain development.**
**(A)** Brain weight at E15.5 (control, N = 8 litters, n = 65 embryos; PE, N = 8 litters, n = 67 embryos), E17.5 (control, N = 8 litters, n = 66 embryos; PE, N = 8 litters, n = 62 embryos), and P0 (control, N = 8 litters, n = 52 pups; PE, N = 8 litters, n = 55 pups). **(B)** Coronal sections of E15.5, E17.5, and P0 offspring cortexes stained with hematoxylin and eosin. Scale bar: 100 $\mu m$ (note different length of scale bars). **(C)** Graph showing the progressive decrease in the radial thickness in rostral to caudal sections of the cerebral cortex in PE-exposed offspring. Different color dots represent E15.5 (green), E17.5 (blue), and P0 (red) control and PE-exposed offspring, respectively (N = 4 litters, n = 8 offspring brains at each time-point for each group). **(D)** Confocal micrographs of coronal sections from E15.5 (left), E17.5 (middle), and P0 (right) control and PE-exposed offspring

for Autism Research Initiative Gene, and found a highly significant degree of overlapping (11 overlapping genes, $P = 0.017$; Fig 2H and Table S3). Validation of two of these overlapping genes (*Grin2a* and *Grin2b*, which encoded NMDA receptors) via qRT-PCR is shown in Fig S3B.

### PE promotes ASD-like behaviors and abnormal synaptic development in adult offspring mice

Next, we assessed whether ASD-like phenotypes exhibited in adult offspring exposed to PE. First, we found no significant differences in the body weight of offspring between the PE and control groups after P3 (Fig 3A) and in the brain weight of offspring at the adult stage (Fig S2F and G), which is in consistent with our previous study on a PE rat model (20). Then, the open-field test (OFT) and the elevated plus maze (EPM) test were performed to evaluate the anxiety levels of the offspring exposed to PE. Both male and female adult offspring exposed to PE exhibited reduced time spent in the center area of the OFT (Fig 3B and C), decreased time spent in the open arms of the EPM (Fig 3D), and increased time spent in the closed arms of the EPM (Fig S4A), compared with offspring born to control mothers. Notably, these behavioral alterations were observed without any changes in total travel distance in the OFT (Fig S4B). These findings indicate that the offspring exposed to PE exhibited reduced exploratory behavior and heightened anxiety levels. Moreover, both adult male and female offspring exposed to PE displayed enhanced marble burying compared with offspring from control mothers (Fig 3E), suggesting that offspring exposed to PE exhibited apparent repetitive behaviors. In the three-chamber test, adult male offspring from PE mothers did not show any preference for interacting with the mouse or the empty cage (Figs 3F and S4C). Unlike adult male offspring exposed to PE, adult female mice exposed to PE exhibited a certain degree of preference for living objects in the three-chamber test (Figs 3F and S4C), implying that male offspring exposed to PE showed social deficits.

Furthermore, RNA-seq of the adult hippocampus in control and PE groups showed that PE exposure led to changes in the expression of 317 genes (209 up-regulated and 108 down-regulated genes) (Figs 3G and S5A and B and Table S4). GO analysis revealed that the most significantly enriched biological processes were "MAPKKK activity, cytoskeletal protein binding and retinoic acid (RA) receptor binding," etc. (Fig S5C and Table S5). KEGG pathway analyses showed that DEGs identified in the hippocampi of mice exposed to PE were enriched in the axonal guidance (Fig S5D and Table S6). qRT-PCR and Western blotting confirmed that the expression of genes associated with ASD or dendritic development, such as *Mnt*, *retinoid X receptor α* (*Rxra*), *E3 ubiquitin* (*Ub*) *ligase* (*Ube3a*), *an AT-rich DNA interacting domain-containing protein* (*Arid1b*) and *Cortistatin* (*Cort*), was changed (43, 44, 45) (Figs 3H and S3C).

Usually, abnormal spine morphology and dendritic development are observed in ASD patients. Thus, we used Golgi staining to investigate spine morphology and dendritic branches. The results showed the spine density of pyramidal neurons in the cortex (Fig S6A and B) and hippocampus (Fig S6C and D) was significantly decreased, as well as the defects of dendritic development (Fig S9) in both adult male and female offspring from PE mothers. Taken together, these data indicated that PE probably promoted both ASD-like behavioral and spine/dendrite abnormalities in offspring born to PE.

## Autism-like signatures in developing brains caused by maternal immune activation via the TNFα/NFκB axis in offspring

### Augmentation of inflammatory cytokine levels in maternal circulation and the TNFα signaling pathway in fetal tissues exposed to PE

Besides hypertension and kidney dysfunction, maternal inflammation is recognized as another core clinical feature of PE in humans (46). In addition, it has been reported that L-NAME–induced PE causes both maternal inflammation and microglial activation in offspring (47). Therefore, we speculated that maternal immunity may be activated by NO blockade, which contributes to ASD symptoms in our PE model. As expected, a mouse Th1/Th2/Th17 array revealed trends toward increases in all the levels of chemokines and cytokines in the sera of PE mothers, compared with control mothers on pregnancy day (PD) 17.5, without alternations in the concentration of total serum protein (Fig 4A). The levels of TNFα were significantly elevated in the PE group compared with the control group (Fig 4A). Concomitantly, the responsiveness to TNFα in placenta (expression of *Tnfaip3* mRNA) and fetal brain (phosphorylation of NFκB) was substantially augmented on E17.5 (Fig 4B–D). However, we observed no significant changes in the levels of chemokines and cytokines in the fetal cortex on E17.5 using the same protein array (Fig 4A).

To investigate the effect of the TNFα/NFκB axis, but not the direct effect of L-NAME (48, 49), on NPC proliferation and dendritic development, we carried out pharmacological experiments in vitro. These experiments showed that TNFα could reduce the proliferation of NPC (Fig 5A–C) and decrease the dendritic length and the number of dendritic branches (Fig 5D–F), as well as the spine density (Fig 5G and H) in cultured neurons. In parallel, Western blotting analysis showed that TNFα, not L-NAME, markedly increased the level of NFκB signaling in NPCs (both p-468 and p536 NFκB) and neurons (only p-536 NFκB) (Fig 5I and J). Inhibition of the NFκB pathway using BAY 11-7082 (50) rescued the adverse effects of TNFα on NPCs and neurons (Fig 5A–H). The sizes of NPC soma were not changed in each group (Fig S7). These results suggested that elevated circulating TNFα levels in PE mothers may activate the NFκB pathway in the fetal brain to reduce the proliferation

---

stained with antibodies against BrdU (green) and p-H3 (red). Hoechst 33342 stains nuclei (blue). The lower panels are enlarged images of the regions outlined by white dotted boxes in the upper panels. Scale bar: 50 $\mu m$. **(E, F)** Quantification of the percentage of BrdU$^+$ (E) and p-H3$^+$ (F) cells per 1,000 $\mu m^2$ (n ≥ 3 brain slices per section from N = 4 litters, n = 8 offspring brains at each time-point for each group). Data are presented as the mean ± s.e.m. NS, $P ≥ 0.05$; *, $P < 0.05$; **, $P < 0.01$; ***, $P < 0.001$, versus control. **(A, C, E, F)** Multiple *t* tests with the Holm–Sidak correction for panels (A, C, E, F) based on the number of litter (N).

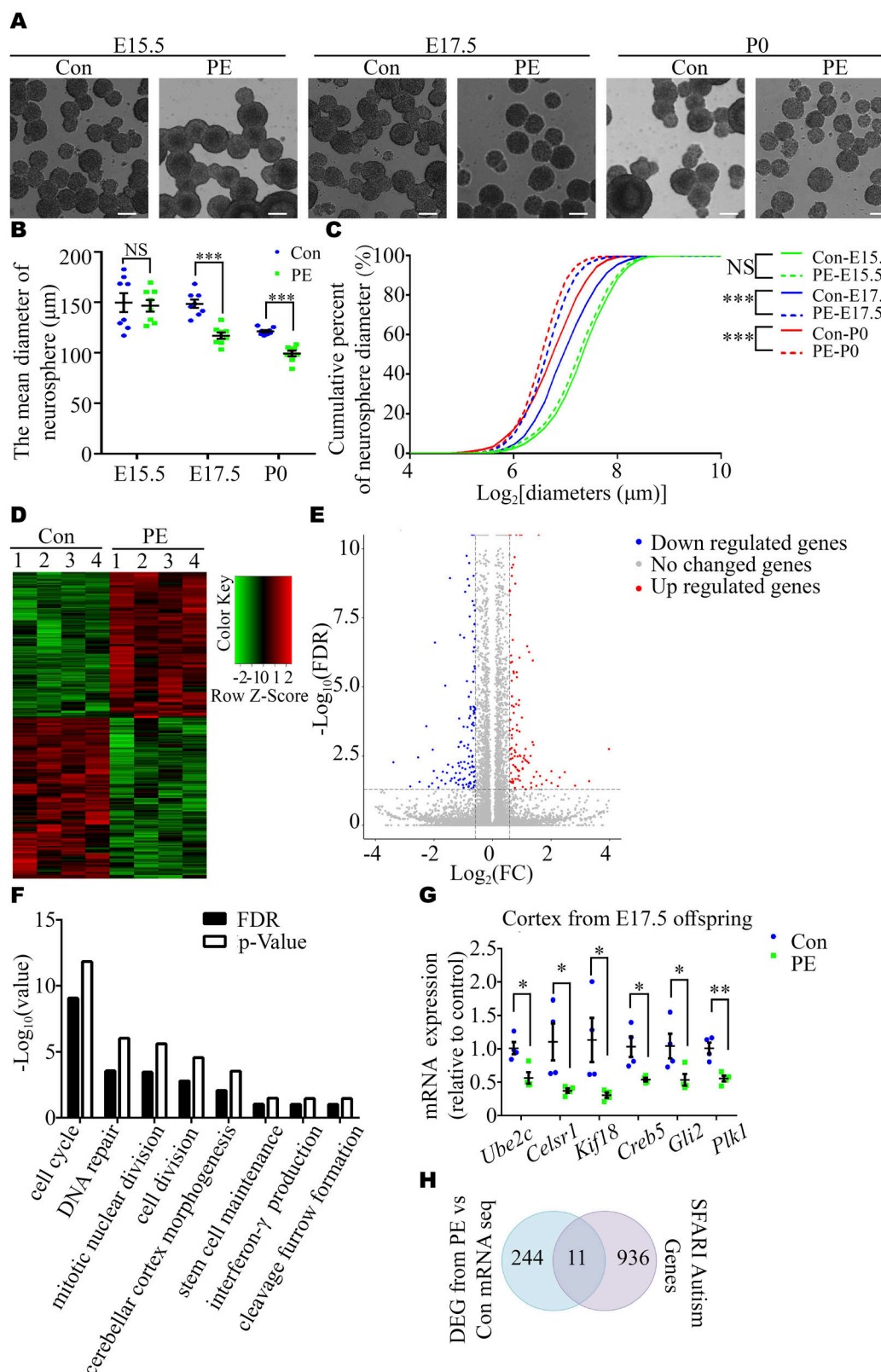

**Figure 2. Inhibition of proliferation in cortex-derived NPC from fetal mice derived from PE mouse model pregnancies.**
**(A)** Representative images of primary cortical neurospheres from each group at E15.5, E17.5, and P0. Scale bars: 100 μm. **(B)** Graph showing the diameter of neurospheres at primary generation at certain time-points (n ≥ 4 cortexes from 4 dams per group at each time-point). **(C)** Cumulative percent of neurosphere diameters (n = 1767/1767, 3071/3473, and 2324/2913 for control (solid lines) and PE groups (dashed lines) at E15.5 (green), E17.5 (blue), and P0 (red). Note the truncated x-axis. **(D, E)** Heatmap (D) and a

capacity of NPCs and disturb the development of neurons in the long term.

### Neutralization of maternal TNFα rescues ASD-like behaviors and synaptic development deficits in adult offspring mice exposed to PE

We reasoned that abnormal embryonic neurogenesis and ASD-related phenotypes in offspring exposed to PE can be rescued by neutralization of maternal TNFα. To test this idea, the pregnancy mice were intraperitoneally injected with mouse TNF antibody on PD16.5, before the abnormal phenotypes in offspring were observed. We found that treatment with TNFα antibody at PD16.5 not only rescued autistic-like behaviors, including anxiety, and repetitive and social behaviors in PE-exposed offspring (Figs 6A–F and S8), but also deficits in the developmental spines on pyramidal neurons in the cortex (Figs S9A and S10A and B) and hippocampus (Figs S9B and S10C and D). Moreover, the phosphorylation level of NFκB in the fetal cortex was restored to normal levels (Fig S11A and B) in PE-exposed offspring treated with TNFα antibody. Notably, treatment with TNFα antibody could not rescue maternal hypertension, or the increase in mortality or decrease in body and brain weight in offspring from PE mothers (Fig S11C–H). No significant changes in water consumption were observed among the groups (Fig S11I). These data suggest TNFα neutralization during pregnancy may benefit the neurodevelopmental outcomes of the offspring exposed to PE in the long term.

## Discussion

Using a mouse model of PE induced by L-NAME via inhibition of NO production, we provide the first experimental evidence that offspring exposed to PE exhibit neurodevelopmental and behavioral symptoms related to ASD. The results confirm the conclusions of accumulating epidemiological studies showing HDP, such as PE, heightens the risk of ASD in offspring (11, 51). Our histological and transcriptomic analyses revealed the decreased neocortex thickness, striatum size, and altered expression of cell cycle–, synapse-, and ASD-associated genes, echoing the alterations in some animal models and patients of ASD (e.g., dendrite dysgenesis and synaptic dysfunction) (52, 53). In addition, immune responses were augmented and TNFα expression was significantly increased in maternal serum in our PE mouse model, which was in line with previous results showing that umbilical and maternal serum levels of TNFα were increased in some PE patients (54). Furthermore, elevated TNFα expression in maternal serum was linked to the activation of NFκB signaling in fetal brains. Neutralization of TNFα in PE mothers improved synaptic development, ameliorated ASD-like behaviors, and reinstated the level of NFκB phosphorylation in fetal cortexes. Definitely, TNFα stimulates NFκB signaling to suppress cell

proliferation in neurospheres and promote early differentiation of NPCs (55, 56), which is in concert with our in vitro experiments. Hypo-activation of NFκB signaling facilitates self-renewal of NPC at the expense of neuron production (57). In addition, overactivation of NFκB induced by cytokines decreases the number of synaptic puncta in vitro (58). Importantly, NFκB signaling can directly alter synaptic gene expression. For instance, NFκB can bind to the promoter of *Grin2a*, which is a risk gene for ASD (59). Reasonably, we speculate that TNFα may directly act on the fetal brain tissues before blood–brain barrier formation, by stimulating the NFκB pathway to disrupt the proliferation of NPC and synaptic development in offspring in our PE model. Furthermore, NFκB signaling can either improve or hinder neurite growth dependent on cellular context via differential phosphorylation sites in a NFκB subunit. For example, enhanced pSer536 NFκB by TNFα inhibits neurite growth in neonatal superior cervical ganglion sympathetic neurons, in contrast to the neurite growth-promoting effects in PC12 cells and sensory neurons (60). Deeper mechanisms should be figured out in the future. For instance, it would be valuable to investigate the effects of different phosphorylation sites of NFκB (such as Ser536 or S468) on neural progenitor cell proliferation or neurite branching. Alternatively, TNFα-induced changes in NFκB signaling could potentially result in alterations in the epigenetic landscape of neural progenitor cells and synaptic development, such as DNA methylation and histone modifications.

It is noteworthy that neutralization of TNFα could not rescue FGR in this model. There are several possibilities that could weaken the benefits of the intervention. The intervention may have been applied too late to restore NPC proliferation. In addition, abnormalities in placental development should be considered. The placenta not only serves as the interface for gas and nutrient exchange and immunological barrier between the mother and fetus, but also secretes hormones to support pregnancy (61). Previous studies indicate that L-NAME treatment in mice can alter placental morphology (62, 63). Starting TNFα antibody treatment on PD16.5 may prevent the restoration of placental function, which could explain why we were unable to improve IUGR in mice. However, we observed that anti-TNFα was able to rescue ASD behaviors in PE-exposed offspring. And we believe that anti-TNFα treatment would be beneficial for both the mother and fetuses. This is because a number of anti-TNF agents, including infliximab, etanercept, adalimumab, certolizumab pegol, and golimumab, are widely used to treat inflammatory diseases such as inflammatory bowel disease, rheumatoid arthritis, ankylosing spondylitis, psoriatic arthritis, and psoriasis (64). Most studies have reported that the use of anti-TNF agents during pregnancy in patients with inflammatory bowel disease, rheumatoid arthritis, ankylosing spondylitis, psoriatic arthritis, and psoriasis does not increase the occurrence of adverse maternal or infant outcomes (65), although there is a higher risk of infections in patients exposed

---

volcano plot (E) showing the 255 DEGs (134 down-regulated and 121 up-regulated DEGs in reference to control samples) in the dorsal cortices of control and PE offspring, as identified by RNA-seq (n = 4 offspring from 4 dams per group). **(F)** GO enrichment analysis of the 134 down-regulated DEGs (FDR<0.05). **(G)** qRT-PCR revealed the decreased expression of six genes associated with the biological functions of cell cycle and stem cell maintenance. **(H)** SFARI genes were significantly overrepresented in the DEG dataset (*P* = 0.017). Data are presented as the mean ± s.e.m. NS, *P* ≥ 0.05; *, *P* < 0.05; **, *P* < 0.01;***, P < 0.001, versus control. **(B, G)** Multiple *t* tests with the Holm–Sidak correction for panels (B, G). **(C)** Non-parametric Kolmogorov–Smirnov test for panel (C).

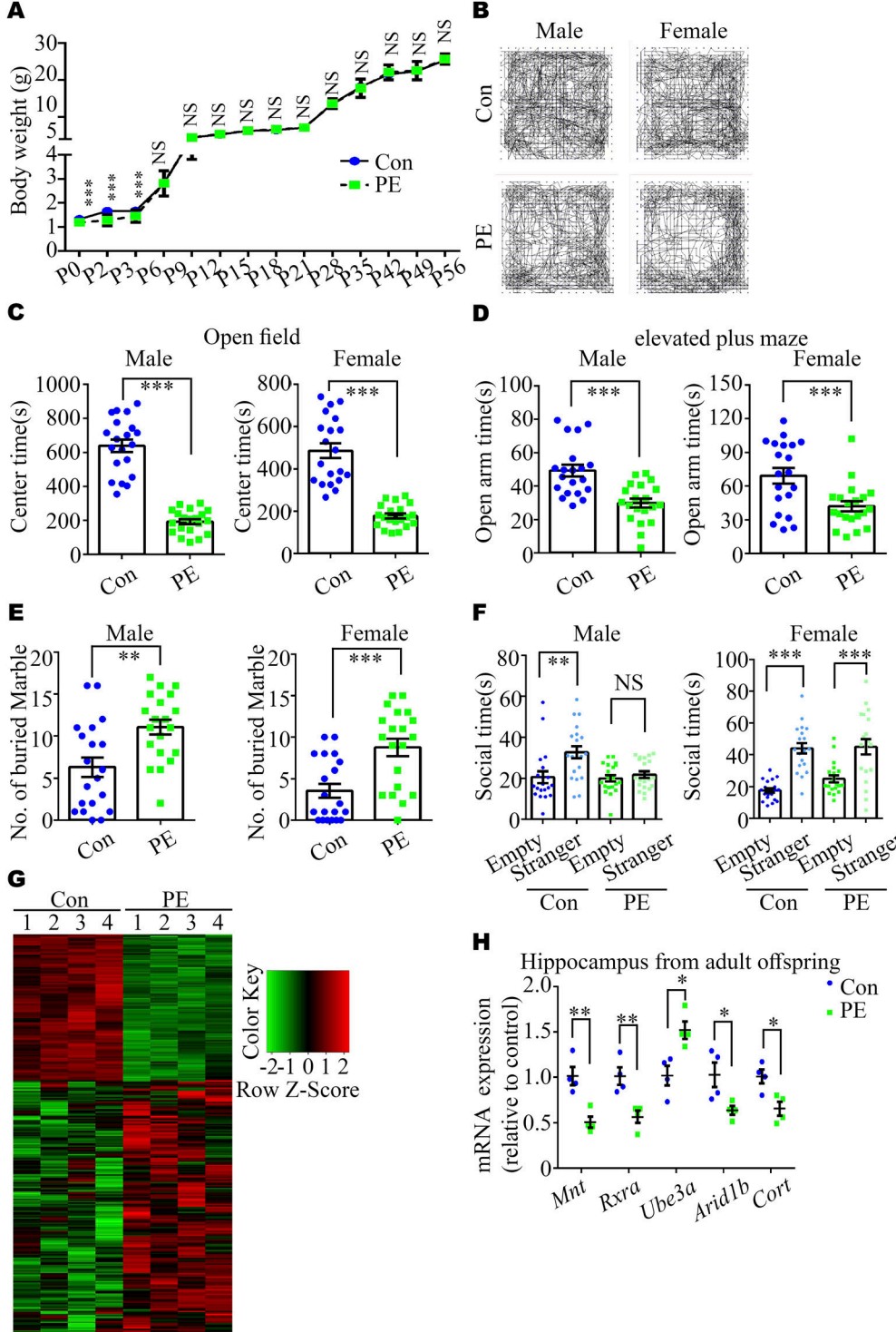

**Figure 3. L-NAME–induced PE increased ASD-like behaviors in adult offspring.**
**(A)** Body weight trajectories of control and PE-exposed offspring (n = 62 control and n = 51–62 PE pups from 8 litters per group at each time-point). **(B)** Representative images of video tracking of control and PE-exposed adult offspring mice in the OFT. **(C, D, E, F)** Time spent in the center area in the OFT (C) and the open arm of EPM (D), the number of buried marbles (E), and the social time for the three-chamber test (F) in the control and PE-exposed adult offspring mice (left panel, male; right panel, female) (n = 20 offspring from 5 litters for each sex per group). **(G)** Heatmap showing 317 (108 down-regulated and 209 up-regulated) DEGs in the hippocampi of control and PE-exposed offspring, as identified by RNA-seq (n = 4 offspring from 4 litters per group). **(H)** qRT-PCR revealed the decreased expression of five genes associated with the etiology of ASD (n = 4 offspring from 4 litters per group). Data are presented as the mean ± s.e.m. NS, $P \geq$ 0.05; *, $P < 0.05$; **, $P < 0.01$; ***, $P < 0.001$, versus control. **(A, H)** Multiple $t$ tests with the Holm–Sidak correction for panels (A, H). Unpaired test for panels (C, D, E). **(F)** One-way ANOVA, Tukey's HSD test for panel (F).

to anti-TNFα (66). The risk-and-benefit balance for both the PE mothers and offspring's health requires further investigation and discussion in future studies. Alternatively, other unknown factors can also disrupt the proliferation of NPC. For example, poor nutrition and/or lower oxygen supplied because of damage to the placenta, which occurs commonly in women with PE, may have contributed to neurogenesis deficits in our PE model (67).

Moreover, our behavioral findings reflected some degree of sex differences in the effect of PE on the development

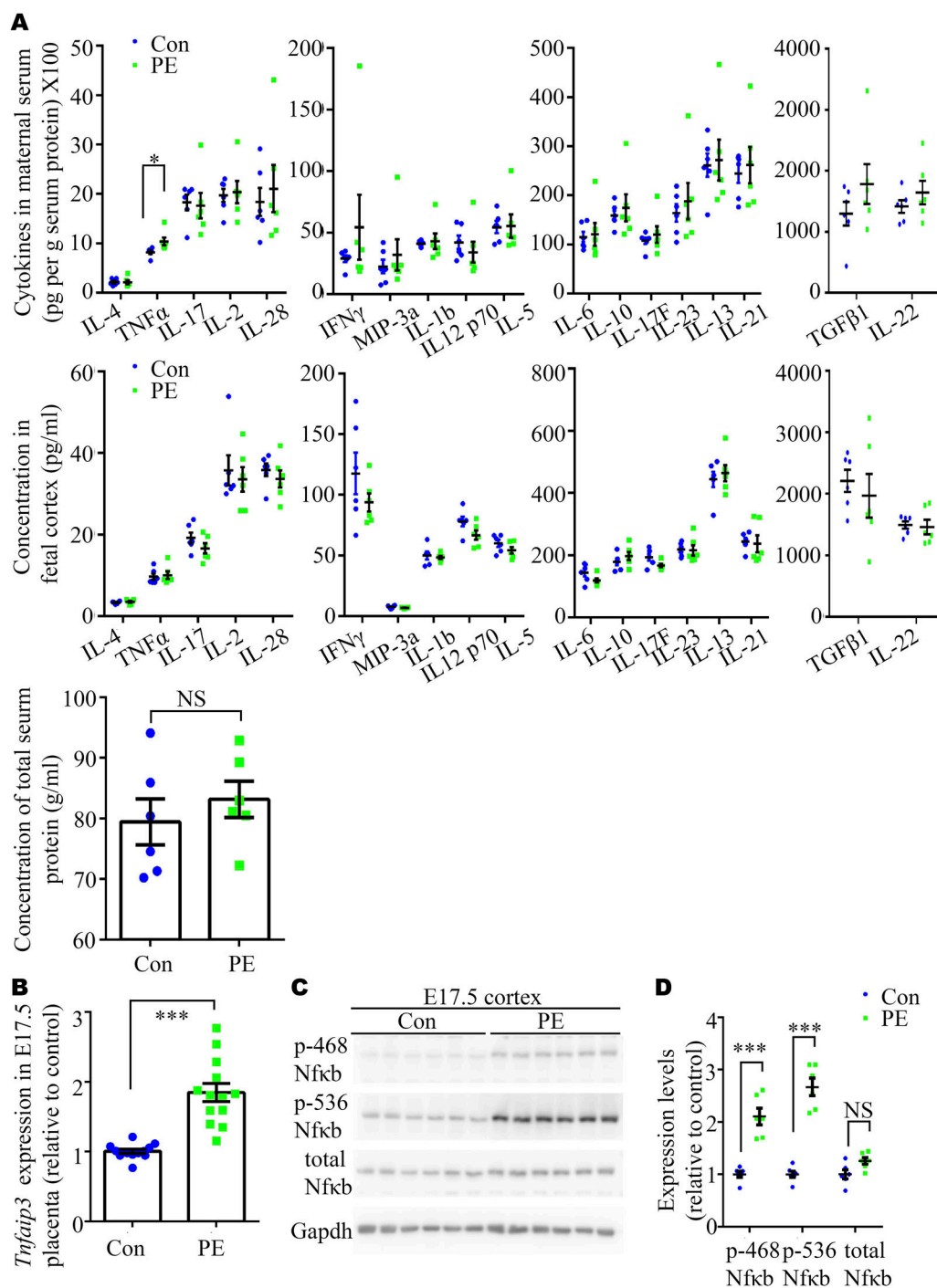

**Figure 4. Expression levels of cytokines in the circulation of PE mother and NFκB phosphorylation in the cortices of their embryos at E17.5.**
**(A)** Mouse Th1/Th2/Th17 array showing multiple cytokines, including IL-4, TNFα, IL-17, IL-2, IL-28, IFN-γ, MIP-3a, IL-1b, IL-1 p70, IL-5, IL-6, IL-10, IL-17F, IL-23, IL-13, IL-21, TNFβ1, and IL-22, in maternal serum (upper panels), fetal cortex (middle panels), and concentration of total serum protein (down panel) at pregnancy day (PD) 17.5 (N = 6 fetuses from six dams for each group). **(B)** qRT-PCR analysis of the expression of Tnfaip3 in the placenta on PD17.5 from control and PE groups. (n = 2–3 placentas per dam, N = 6 dams per group). **(C, D)** Representative Western blotting images (C) and a graph (D) showing overactivation of NFκB in the cortices of embryos from PE mothers on E17.5 (n = 6 fetal cortexes from three dams per group). Data are presented as the mean ± s.e.m. NS, $P \geq 0.05$; *, $P < 0.05$; ***, $P < 0.001$, versus control. **(A, D)** Multiple $t$ tests with the Holm–Sidak correction for panels (A, D). **(B)** Unpaired, two-tailed test for panel (B).

of ASD in offspring. The male offspring from PE mothers exhibited all core ASD behaviors, whereas female offspring exposed to PE showed no significant alterations in social behaviors, except increased anxiety levels and repetitive behaviors. Prior works have suggested that female hormones or placental sex differences might influence the development and

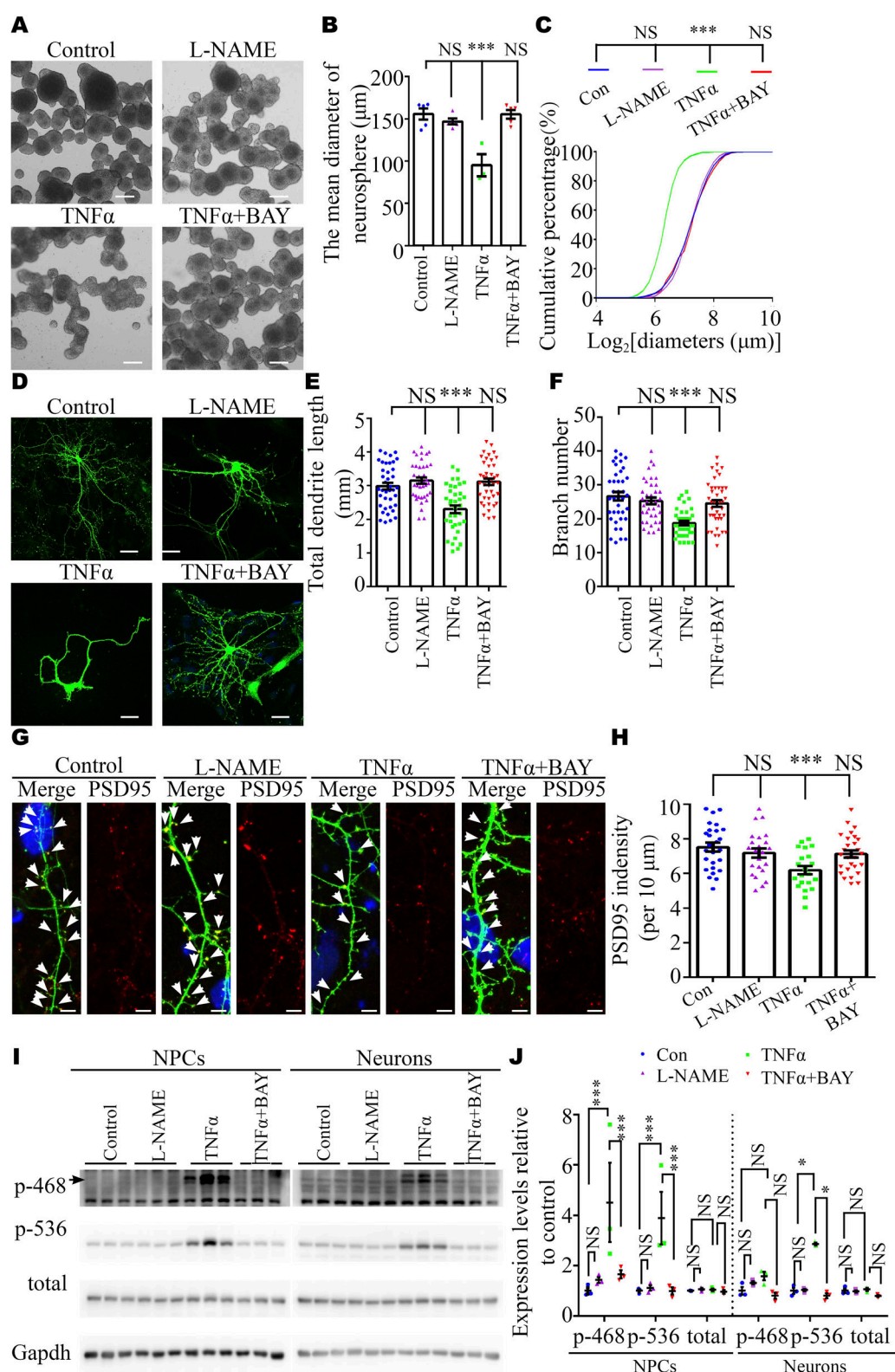

**Figure 5. TNFα, but not L-NAME, inhibited the self-renewal of NPC and neurite and synaptic development in vitro.**
**(A)** Representative images of primary neurospheres from E17.5 embryos treated with L-NAME, TNFα (10 ng/ml), or TNFα (10 ng/ml) + BAY 11-7082 (10 μM). Scale bar: 100 μm. **(B)** Graph showing the diameter of neurospheres (n = 6, 6, 3, and 6 replicates for control, L-NAME, TNFα, and TNFα+ BAY 11-7082 groups, respectively). **(C)** Cumulative percentages of neurospheres of various diameters (n = 438–1250 neurospheres). Note the truncated x-axis. **(D)** Representative images of immunostaining of GFP in cultured E17.5 cortical neurons. Scale bar: 20 μm. **(D, E, F)** Analysis of dendritic length and branch number of neurons in panel (D) (n = 39, 37, 38, and 42 neurons in

function of the brain (68). Whether these factors that govern social interaction are affected by PE in a sex-specific manner should be identified in future work.

Notably, different cytokines elevated during pregnancy may affect embryos through various pathways to cause ASD-like phenotypes in offspring. For example, in mice, chronic IL-17a treatment during pregnancy alters cortical development in male but not female offspring and behaviors in the adult stage (69). Moreover, FGR caused by chronic IL-17a treatment is symmetrical and persistent, different from the asymmetrical, brain-sparing growth restriction observed in PE patients (69, 70) and our model. The number of neurites, total neurite length and number of branches are increased in primary neurons treated with serum containing high levels of IL-6 from patients with PE compared with those treated with control serum (71). Contrary to these results, our findings showed relatively asymmetrical growth restriction between body and brain (data not shown), and transient changes after delivery, as well as non–sex-specific changes in anxious and repetitive behaviors in adulthood in our PE model. Thus, whether maternal immune activation occurs and which inflammatory factors are specifically altered in PE patients and animal models should be emphasized in future research.

Furthermore, we did not account for the potential postnatal influences of maternal inflammation. A previous study reported that maternal hematopoietic TNF$\alpha$ can affect behaviors in offspring via milk chemokines (72). Therefore, cross-fostering by control and PE dams may be useful to distinguish prenatal and postnatal impacts on adult offspring. In addition, impaired spatial learning and memory were observed in the offspring from PE groups in both this study (Fig S4D and E) and a previously published rat PE model (20), whereas hippocampal-dependent spatial memory is augmented in the offspring of TNF-null dams. In clinical studies, other neurobiological deficits in working memory, oculomotor control (73), and verbal ability (74) have been observed in children born to mother with PE (72). Thus, whether TNF$\alpha$ and its related pathways are involved in hippocampal-dependent behaviors in addition to autism-specific behavior should be investigated in future work.

In addition, our transcriptomic analysis of the hippocampi of adult male mice revealed PE had long-term detrimental effects on offspring, likely by altering the axonal guidance pathway and/or RA signaling. In parallel, these results provided several novel potential targets to ameliorate ASD-like behaviors in offspring of PE model mice. Hu Ronggui and his colleagues reported that hyperactivity of Ube3a causes 1–3% of ASD cases worldwide through degrading Aldh1a2, a rate-limiting enzyme of RA synthesis. RA supplementation significantly alleviates ASD-like behaviors in *Ube3a*-overexpressing mice (43). Intriguingly, the

expression of RA response genes including *Rxra* and secretory carrier–associated membrane protein 3 (*Scamp3*) was down-regulated, whereas the expression of Ub-mediated proteolysis-related genes including *Ube3a* was up-regulated in our study. These observations raise the possibility that a reduction in RA signaling and Ube3a hyperactivity are additional mechanisms that underlie ASD phenotypes in adult offspring exposed to PE. Thus, augmentation of RA signaling may alleviate the ASD-like symptoms of the offspring born to mothers with PE during the postnatal period. Moreover, the expression of *Arid1b*, considered as an ASD risk gene (44), is down-regulated in the hippocampi of offspring from PE mothers in this study. Taken together, the results suggest that PE dramatically and persistently changes the expression of genes in the brains of PE-exposed offspring.

Finally, it is important to acknowledge the limitations of our current study. One such limitation is the potential mild chronic stress resulting from the handling and restraint required when measuring blood pressure in mice. Although we took several steps to mitigate this effect—including a 3-d training period before L-NAME treatment, consistent measurement times, gentle handling of the animals, and conducting measurements in a quiet room—it is possible that this stressor could still interact with L-NAME in pregnancy. In addition, it is worth noting that our L-NAME–induced PE animal model only partially recapitulates certain features of PE in humans, such as SBP, proteinuria, and IUGR. Therefore, it is crucial to further investigate the effects of PE on ASD using other PE models.

Nevertheless, the salient finding of this study is that elevated TNF$\alpha$ levels in maternal circulation contribute to the occurrence of ASD-like phenotypes in the offspring exposed to PE mothers by activation of NF$\kappa$B signaling. Neutralization of TNF$\alpha$ restores NF$\kappa$B signaling activation, improves the neurodevelopmental outcomes, and ameliorates ASD-like behaviors of the offspring from PE mothers. Furthermore, this study highlights the contribution of maternal inflammation to detrimental neurodevelopmental consequences in a PE model. In addition, this work provides several new perspectives on ASD etiology and valuable targets for the prevention and treatment of ASD and other related psychopathologies in offspring born to PE mothers.

# Materials and Methods

### Animals

All C57BL/6J mice were purchased from SLAC Laboratory Animal Co., Ltd and housed in a standard SPF room with free access to food and water, under a 12:12-h light: dark cycle at 22°C ± 1°C, 40–60%

---

control, L-NAME, TNF$\alpha$, and TNF$\alpha$ BAY 11–7082 groups, respectively; N = 4 independent experiments). **(G)** Representative images of immunostaining of GFP (green) and PSD95 (red) in cultured E17.5 cortical neurons. Scale bar: 5 $\mu$m. **(G, H)** Quantitative analysis of PSD95 density in panel (G) (counted neuron numbers, n = 26, 23, 21, and 28 neurons in control, L-NAME, TNF$\alpha$, and TNF$\alpha$+ BAY 11-7082 groups, respectively; N = 4 independent experiments). **(I, J)** Representative Western blotting images (I) and a graph (J) showing phosphorylated NF$\kappa$B expression in NPCs (left panel) and neurons (right panel) (N = 3 independent experiments). Data are presented as the mean ± s.e.m. NS, $P \geq 0.05$; **, $P < 0.01$; ***, $P < 0.001$, versus control. **(B, C, E, F, H, J)** One-way ANOVA, Tukey's HSD for panels (B, E, F, H); non-parametric Kolmogorov–Smirnov test for panel (C); two-way ANOVA, Tukey's HSD for panel (J).

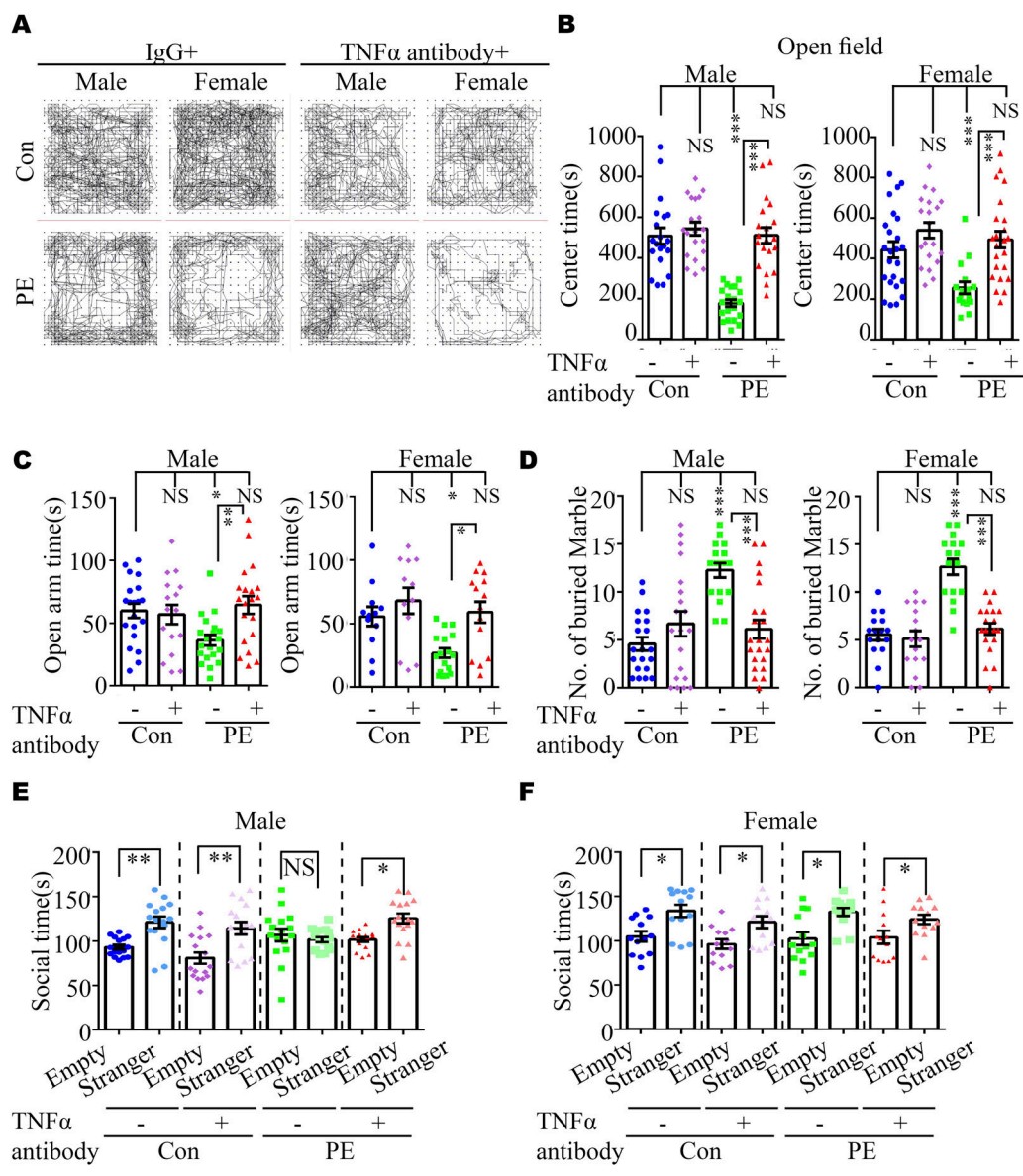

**Figure 6. Neutralization of TNFα in maternal serum ameliorated ASD-like phenotypes and suppressed NFκB signaling in the PE-exposed offspring.**
**(A)** Video tracking of adult offspring mice in control+IgG, control+TNFα antibody, PE+IgG, and PE+TNFα antibody groups in the OFT. **(B, C, D, E, F)** Time spent in the center area in the OFT (B) (n = 20 male offspring from 5 litters per group and n = 24, 20, 24, and 16 female offspring from 4–6 litters for control+IgG, control+TNFα antibody, PE+IgG, and PE+TNFα antibody groups, respectively) and the open arm of EPM (C) (n = 20, 16, 20, and 20 male offspring from 4–5 litters and 20, 16, 20, and 16 female offspring from 4–5 litters for control+IgG, control+TNFα antibody, PE+IgG, and PE+TNFα antibody groups, respectively); the number of buried marbles (D) (n = 20, 20, 16, and 22 male offspring from 4–5 litters and 16, 16, 16, and 20 female offspring from 4–5 litters for control+IgG, control+TNFα antibody, PE+IgG, and PE+TNFα antibody groups, respectively) (left panel, male; right panel, female); and the social time for the three-chamber test (E, F) (n = 16 offspring per sex per group from 4 litters) for offspring, derived from control+IgG, control+TNFα antibody, PE+IgG, and PE+TNFα antibody groups, respectively. Data are presented as the mean ± s.e.m. NS, $P \geq 0.05$; *, $P < 0.05$; **, $P < 0.01$; ***, $P < 0.001$, versus control. **(B, C, D, E, F)** One-way ANOVA, Tukey's HSD for panels (B, C, D, E, F).

humidity. 8-wk-old dams and sires were used for breeding. The day on which the vaginal plug detected was considered day 0.5 of pregnancy. Pregnant dams were given drinking pure water (control group) or water with 1 mg/ml L-NAME (N5751; Sigma-Aldrich) (PE group) from PD12.5 to the delivery day. The concentration of L-NAME was determined by previous studies (20, 75). For the rescue assay, dams from the control and PE groups were intraperitoneally administered normal rabbit IgG (1.5 μg/g body weight) (2729; Cell Signaling Technology) and TNFα antibody (1.5 μg/g body weight)

(500-P64; PeproTech), respectively (76). Successful modeling was confirmed via SBP measured with BP-2000 Blood Pressure Analysis System (Visitech Systems Inc.), and 24-h urinary protein levels were measured using a BCA protein assay kit (23227; Thermo Fisher Scientific Inc.). Offspring from control and PE groups were raised by their respective mothers, and their weight trajectories were monitored using an electric scale. Mortality was calculated as A(the number of offspring observed at P0)−B(the number of offspring observed at P2)/ A(the number of offspring observed at P0).

**Life Science Alliance**

## Ethics approval statement

These animal experiments were approved by and conducted in accordance with the Shanghai Jiao Tong University Animal Ethics Committee (approval no. GKLW2016-31) and conducted in accordance with the guidelines.

## ASD behavioral tests

Male and female mice aged greater than 8 wk were used in all behavioral tests. Before the beginning of the behavioral tests, the animals were handled for 3 d and put in the experimental room for 30 min to allow them to adapt to the environment. The apparatuses used in this study were washed with 70% ethanol and water before each trial. In all experiments except the marble burying test, the behavioral trajectories of the mice were recorded and automatically calculated by a Noldus video analysis system (Noldus Information Technology) as described previously (77). The detailed methods for ASD behavioral tests are shown in Supplemental Data 1.

Briefly, the OFT was used to assess the motor function and anxiety level of each mouse. The EPM test was used to evaluate the anxiety level of each mouse. The marble burying test was used to assess the repetitive behaviors of the mice. The three-chamber test was used to assess the social behavior of the mice.

## Primary cultures of hippocampal neurons

Primary hippocampal neurons were dissected from a E17.5 mouse of either sex from each group. Following a previously reported method (78), the neurons were cultured in 0.5 ml/well Neurobasal medium (21103-049; Gibco) supplemented with 0.2% B27 (17504-044; Gibco) and 2 mM GlutaMAX-I (35050-061; Gibco) on 12-mm $\varphi$ glass slides at a density of 100,000 cells/cm$^2$. The neurons were treated with 10 $\mu$g/ml TNF$\alpha$, 1 $\mu$g /ml L-NAME, and 10 $\mu$M BAY 11-7082 (B5556; Sigma-Aldrich) after 3 days in vitro (DIV3).

## Calcium phosphate transfection

The neurons were transfected with a GFP expression vector (pEGFP-N1) with a calcium phosphate transfection kit (C0508; Beyotime) according to the manufacturer's instructions at 5 d after seeding. The detailed method is shown in Supplemental Data 1.

## Golgi staining

Age- and gender-matched mice from each group were deeply anesthetized with 0.14 g/kg sodium pentobarbital. Then, the brains were stained using FD Rapid GolgiStain Kit (PK401; FD Neuro-Technologies). The detailed instructions are listed in Supplemental Data 1.

## Neurosphere and neuroprogenitor cell culture

The cerebral cortices of E15.5 and E17.5 mouse embryos and P0 pups were prepared as previously described (79). Single dissociated cells were cultured in a serum-free medium consisting of high-glucose DMEM, L-glutamine, Na-pyruvate, B-27, and N2 supplemented with EGF and FGF2 (10 ng/ml each) in uncoated six-well plates for neurosphere or poly-D-lysine (P6407; Sigma-Aldrich)–coated 12-mm $\varphi$ glass slides in a 24-well plate. Single dissociated cells from each embryo or pup were collected in duplicate. Images of neurospheres in each well were acquired by microscopy after 6 d. The diameter of the neurospheres was measured using Image-Pro Plus software, and only neurospheres larger than 20 $\mu$m in diameter were assessed. All embryos and pups from at least 3 litters per group were used in the experiments. The neurospheres were treated with 10 $\mu$g/ml TNF$\alpha$, 1 $\mu$g /ml L-NAME, and 10 $\mu$M BAY 11-7082 (B5556; Sigma-Aldrich) after DIV3 days. As shown in Fig S7A, neurospheres were first infected with a lentivirus packaged with a EF1$\alpha$-driven GFP expression vector (L1020; Walgen, Inc.) for 6 h and dissociated with Accutase (A1110501; Thermo Fisher Scientific) at 37°C and 5% CO$_2$ for 15 min after aforementioned drug treatment. The single dissociated cells were then seeded onto poly-D-lysine–coated 12-mm $\varphi$ glass slides in a 24-well plate and cultured in a culture medium with corresponding drugs for 24 h. Then, the cells were fixed and subjected to immunofluorescence (IF) procedures.

## Protein array

According to the manufacturer's instructions, the sera of pregnant mice and corresponding embryonic cortices were assessed on PD17.5 with mouse Th1/Th2/Th17 arrays (QAM-TH17-1; RayBiotech) by BioTNT Company. The fluorescence signals were visualized with a laser scanner in a Cy3 channel. The total serum protein was measured using the biuret method with Protein Content Assay Kit (BC3185; Solarbio) according to the provided instructions.

## Hematoxylin and eosin staining

Following the manufacturer's instructions, the brain slices were stained with a hematoxylin and eosin kit (C0105; Beyotime).

## IF

Pregnant mice on PD15.5 and PD17.5 and pup at P0 day were injected intraperitoneally with BrdU at 50 mg/kg body weight. The animals were deeply anesthetized with 0.14 g/kg sodium pentobarbital. Then, the brains were treated as the traditional procedures of IF as previously described (20). For adhered cells, the cultured neurons and neuroprogenitors were fixed with 4% PFA for 10 min at room temperature following standard protocols as described previously (80). The detailed method is described in Supplemental Data 1. The primary and secondary antibodies used in the present study are listed in Table S7.

## Western blots

Protein was extracted from the dorsal cortices of E17.5 embryos with lysis buffer supplemented with protease and phosphatase inhibitors (5892791001; Roche) as previously described (81). The primary and secondary antibodies used in the present study are listed in Table S7.

## qRT-PCR

Total RNA was extracted from the dorsal cortices of E17.5 embryos (either sex) and the hippocampi of adult male mice (4–5 offspring per group) using TRIzol (15596026; Invitrogen). The concentration of RNA was determined (NanoDrop Spectrophotometer, Thermo Fisher Scientific), and equivalent amounts of RNA were reverse-transcribed into first-strand cDNA using PrimeScript RT reagent kits (RR047A; TaKaRa). qRT-PCR was carried out with SYBR Green Kits (DRR063A; TaKaRa) in triplicate, calculated from average $C_t$ values normalized to *Gapdh* as an internal control. The primer sequences used in this assay are listed in Table S8.

## RNA-seq

RNA extracted from the dorsal cortices of E17.5 embryos and the hippocampus of adult male mice with RNA integrity number >9.0 was considered acceptable for cDNA library construction. cDNA libraries were constructed for each RNA sample using TruSeq Stranded mRNA Library Prep Kit (Illumina, Inc.) according to the manufacturer's instructions. And the qualities of the libraries were controlled with an Agilent 2200 device and sequenced by HiSeq X (Illumina) on a 150-bp paired-end run. Then, the clean reads were obtained and aligned to the mouse genome (mm10, NCBI) using Hisat2 (82). HTSeq was applied to obtain gene counts, and gene expression was determined using the RPKM method (83).

## Statistical analyses

Statistical results were performed using GraphPad Prism 6 (SCR_002798; GraphPad Software, Inc.). Normality and equal variances between groups were assessed using a Kolmogorov–Smirnov test. When normality and equal variance were achieved between groups, the comparisons were analyzed by a t test, or multiple *t* tests with the Holm–Sidak correction, or one-way ANOVA with Tukey's multiple comparisons test, or two-way ANOVA with Tukey's multiple comparisons test. If not, a Kruskal–Wallis test with Dunn's correction or a Mann–Whitney *U* test, or a Wilcoxon signed-rank test was carried out. The detailed statistical methods are noted in both the corresponding figure legends and Supplemental Data 1. Cumulative plots of neurosphere diameters were analyzed using the Kolmogorov–Smirnov test. The values of neurosphere diameter were first transformed by $\log_2$ and then applied in cumulative distribution analysis, according to a previous study (84). For the analysis of DEGs, the EBSeq algorithm (85) was applied to filter DEGs, after the significant analysis, *P*-value, and false discovery rates (FDRs) were calculated. The following criteria were used: (i) $\log_2$(fold change) >0.585 or <−0.585 for E17.5 RNA-seq and >0.263 or <−0.263 for adult hippocampus RNA-seq; and (ii) FDR<0.05 (86). GO analysis was performed to facilitate elucidating the biological implications of the DEGs in the experiment. DEGs in the cortices of E17.5 embryos exposed to PE were compared with the genes in the Human Gene module of Simons Foundation for Autism Research Initiative using enrichment analysis via phyper function in R.

## Data Availability

All data needed to evaluate the conclusions are present in the study and the Supplementary Materials. The raw RNA-seq data were uploaded to Gene Expression Omnibus (GSE167193 and GSE167194).

## Supplementary Information

## Acknowledgements

This work was funded by the National Natural Science Foundation of China (Grant nos. 81501281, 82071730, and 81801469), the National Key Research and Development Program of China (2021YFC2700701 to X Lin), and the Clinical Research Plan of SHDC (SHDC12019X17 to X Lin). The authors also thank the RNA-seq service from NovelBio Company.

### Author Contributions

X Liu: conceptualization, data curation, funding acquisition, and writing—original draft.
H Liu: data curation and methodology.
N Gu: validation and methodology.
J Pei: validation and methodology.
X Lin: conceptualization, supervision, funding acquisition, project administration, and writing—review and editing.
W Zhao: conceptualization, supervision, funding acquisition, and writing—review and editing.

### Conflict of Interest Statement

The authors declare that they have no conflict of interest.

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
