## [Reviewer comments · Life Science Alliance]

Life Science Alliance

Preeclampsia promotes autism in offspring via maternal inflammation and fetal NF κ B signaling

Xueyuan Liu, Haiyan Liu, Nihao Gu, Jiangnan Pei, Xianhua Lin, and Wenlong Zhao

DOI: <https://doi.org/10.26508/lsa.202301957>

Corresponding author(s): Dr. Wenlong Zhao (Rutgers, The State University of New Jersey)

Review Timeline:

Submission Date:	2023-01-27
Editorial Decision:	2023-02-17
Revision Received:	2023-04-25
Editorial Decision:	2023-05-23
Revision Received:	2023-05-28
Accepted:	2023-05-30

Scientific Editor: Novella Guidi

Transaction Report:

February 17, 2023

Re: Life Science Alliance manuscript #LSA-2023-01957

Dr. Wenlong Zhao
Rutgers, The State University of New Jersey
170 Frelinghuysen Rd
Piscataway 08854

Dear Dr. Zhao,

Thank you for submitting your manuscript entitled "Preeclampsia promotes autism in offspring via maternal inflammation and fetal NFκB signaling" to Life Science Alliance. The manuscript was assessed by expert reviewers, whose comments are appended to this letter. We invite you to submit a revised manuscript addressing the Reviewer comments.

Thank you for this interesting contribution to Life Science Alliance. We are looking forward to receiving your revised manuscript.

Sincerely,

B. MANUSCRIPT ORGANIZATION AND FORMATTING:

Reviewer #1 (Comments to the Authors (Required)):

In the manuscript titled "Preeclampsia promotes autism in offspring via maternal inflammation and fetal NF κ B signaling," the authors investigated the association between Preeclampsia (PE) and autism spectrum disorder (ASD) potential in the offspring of a rodent model. This study showed that PE-exposed offspring exhibit characteristics of ASD, including neurodevelopment deficiency and behavioral abnormalities. They also identified the levels of inflammatory cytokines, especially TNF α , in the maternal serum and elevated NF κ B signaling in the fetal cortex, contributing to the deficits in neural progenitor cell (NPC) proliferation and synaptic development. Most interestingly, neutralizing TNF α in vitro and in PE mothers improved synaptic formation and cortical development, further ameliorating ASD-like behaviors in the offspring.

This manuscript presents some solid data and fantastic results, but minor improvements could be made in further revision (listed below).

1. Fig. 1: The authors found a dramatic decrease in the cortex length of PE-exposed pups compared to control cohorts. They also suggested the hippocampus region was damaged in the next couple of figures. I doubt other brain regions may also be affected in these pups. Did they perform a systematic examination across the entire brain? Some previous reports suggested that deficits from the striatum and cerebellum may contribute to ASD. How about these regions?
2. Still in Fig. 1: Is the decrease in cortical length due to the reduction in total cell numbers, or just only loss in the proliferating progenitors? How about the miss-alignment of newborn neurons (affecting neuronal migration)? The author should be able to show the analysis, or at least consider the above possibilities in the context.
3. Fig. 2: the authors identified a proliferation reduction in the PE-progenitors both in vivo and in vitro. What specific cell type is that? Radial glia, Intermediate progenitor cells (IPCs), neuronal precursors, or glia precursors? Is there any reduction in the total neuron number or glia?
4. I am not sure whether the authors examine adult PE-exposed mice brains. Is the PE brain normal? Size? Weight?
5. Fig 3: since the hippocampus region was also affected in PE mice, is there any deficit in hippocampus-related behavior? Learning disability?
6. Fig. 5 and Fig. S5: the author should state which cortical layer of neurons was analyzed in this research.

Although there are some limitations to this study, I still believe it is a sound and well-prepared piece of work. The conclusions are solid and of significant medical interest, which composed a very important piece of clinical research. Overall, I suggest a "minor revision" opportunity for this manuscript.

Update (2/15/2023):

Thank you very much for letting us see other reviewer's opinions! I think it's a good idea!

I agreed with reviewer #2 that, the language of this manuscript needs to be carefully polished. Also, the methods part needs more details as reviewer #2 suggested.

In general, I still insist on my previous opinion that I would like to recommend this manuscript to the journal. It's exciting research!

Reviewer #2 (Comments to the Authors (Required)):

Thank you for the opportunity to review this interesting paper, which presents findings related to the role of an animal model of preeclampsia (chronic L-NAME) on in utero offspring brain programming relevant to ASD. This work constitutes a novel contribution to the field and will advance our knowledge of inflammatory mechanisms of neurodevelopmental programming by gestational diseases and exposures which impact placental health and function.

Despite enthusiasm, some details are absent from the report and further rationale is required. Methods should be fully explained throughout, and some additional analyses considered, as described below. Additionally, please consider that some assessments (embryonic) might consider the biological replicate to be the litter rather than the individual animal. Language should also be revised throughout for clarity.

The data are supportive of the conclusion that maternal L-NAME leads to ASD-like phenotypes in offspring, and that some but not all of these are dependent on TNF α signaling.

Some specific comments and suggestions are listed:

Abstract:

The authors state "animal model and underlying mechanisms were not reported yet, hindering the development of effective therapeutic approaches." However, multiple studies have focused on the developing brain impacts of preeclampsia (

"characteristics of ASD" should be revised to "ASD-like phenotypes" or "ASD-like characteristics"

Introduction:

Prevalence data for ASD should be updated to more recent figures (e.g., 1:44 in the US, CDC), as rates change yearly and have risen since 2014.

Line 36: The 17-50% contribution by environmental drivers of ASD should be cited.

Lines 38, 48: You may also wish to cite reviews on the topic of PE impacts on neurodevelopmental disabilities and ASD (PMID: 35872512, PMID: 32209456)

Line 57-58 requires a citation: " which results in preterm birth and escalates the risk for aberrant neurodevelopment in offspring."

Discussion of the existing literature linking animal models of PE to ASD-like phenotypes and neurodevelopmental deficits is lacking some references (e.g., line 64-65: lacks studies on PE animal models to mimicked key characteristics of ASD in offspring exposed to PE). Consider referencing: PMIDs 33510137, 30683649, 25575681. Many of these are summarized in PMID: 32209456.

lines 67-68: Please specify PE-endophenotypes recapitulated by the chronic L-NAME model

The rationale for focusing on TNF α should be more thoroughly described. For example, are other disorders of elevated TNF α in pregnancy linked to ASD?

Methods:

How old were dams and sires at breeding?

Was water consumption altered in the L-NAME group?

What concentration of IgG was administered?

There is some concern with the use of photoplethysmography with dark skin/hair (e.g., C57Bl/6 animals). This should be briefly mentioned/discussed.

Handling and restraint stress related to photoplethysmography may also be considered a mild chronic stressor, which may interact with L-NAME in pregnancy. This should also be discussed.

The breeding scheme is unclear. The authors write "Offspring from control and PE groups were bred with their mothers and weighted using an electric scale to trace their weight trajectories." Does this mean that the ordered animals were not used in experiments, but rather their progeny, who were exposed to L-NAME in utero and then when they themselves were pregnant? This would be an inter-generational model and different from what is described elsewhere in the paper.

Were litters culled? This may be a source of variance in rearing experience of offspring. What was the average litter size in each group at behavioral testing?

For the three-chamber social task, stranger mice should ideally be sex, age, and size-matched.

What software was used for each task and for behavioral quantification? If scoring was manual, was it repeated and tested for inter-rater reliability?

What was the standard housing bedding material for these animals? Consider thoroughly reporting all rearing/housing conditions and manipulations per the guidelines outlined in PMID: 30188509. You may attach their helpful template table as a supplement to your paper.

Supplement IF section: What is meant by "enough PBS and 4% PFA" ?

How many dams were bred per condition? Were resorptions detected (at E15, E17)? Why were E15.5 and E17.5 selected as timepoints for embryonic brain analyses? Some rationale would be helpful.

Were cytokine levels normalized to total protein in serum?

Results:

Average litter-wise brain weight and cortical thickness data should be provided (with averages per litter within each condition), as the litter may be considered the biological replicate.

Line 83: "12 days after pregnancy" suggests a single post-partum manipulation. This should be revised to "at gestational day (GD) 12 through..." or similar.

Consider breaking up the results section into further subsections for improved clarity and flow

Lines 185-187: If these samples were paired, were placenta Tnfaip3 and brain NFkB correlated across PE or control (or all) fetuses?

Line 189: E17.5 cortical array data should be shown in possible. Were these below limit of detection, or were they truly unchanged?

Was NPC soma size/volume changed by NFkB manipulations?

Line 215: "stored" should be changed to "restored"

EPM: Open/Closed arm time should be calculated and reported

Social: A sociability quotient (social-nonsocial/total) might be also be calculated and reported to normalize social time to time spent with the empty cup

Open field: total distance traveled should be shown in figures.

Figures:

Fig 2C: "Percentage" is misspelled. This should be specified as "percent of X"

Fig 2E: down and up regulated in reference to what? This should be specified in the legend

Discussion:

The TNFa antibody rescue experiment occurs at PD16.5, which is well after placentation processes have occurred. Placentation abnormalities are often credited with driving many of the physiologic disruptions in preeclampsia. How might the authors explain this limitation or caveat in their interpretation?

Line 254: "Deeper mechanisms should be figured out in future" is a non-specific statement. What might some of these mechanisms for future consideration be? Several should be suggested.

The clinical literature on use of anti-TNFa agents should be briefly discussed, particularly as it pertains to impacts on gestation and offspring brain outcomes (e.g., PMID: 34489011)

Do you predict that your results pertain to a subset of ASD patients, for whom prenatal inflammation is a driving pathoetiologic factor? ASD is a highly variable, spectral disorder with many different potential causes/influences. Does your work highlight a need for improved individualized treatments/preventions in ASD?

Reply to Reviewers

Please see Reviewers' comments below in **black** and also our reply **marked in yellow**.

Reviewer #1 (Comments to the Authors (Required)):

In the manuscript titled "Preeclampsia promotes autism in offspring via maternal inflammation and fetal NF κ B signaling," the authors investigated the association between Preeclampsia (PE) and autism spectrum disorder (ASD) potential in the offspring of a rodent model. This study showed that PE-exposed offspring exhibit characteristics of ASD, including neurodevelopment deficiency and behavioral abnormalities. They also identified the levels of inflammatory cytokines, especially TNF α , in the maternal serum and elevated NF κ B signaling in the fetal cortex, contributing to the deficits in neural progenitor cell (NPC) proliferation and synaptic development. Most interestingly, neutralizing TNF α in vitro and in PE mothers improved synaptic formation and cortical development, further ameliorating ASD-like behaviors in the offspring.

This manuscript presents some solid data and fantastic results, but minor improvements could be made in further revision (listed below).

Question 1: Fig. 1: The authors found a dramatic decrease in the cortex length of PE-exposed pups compared to control cohorts. They also suggested the hippocampus region was damaged in the next couple of figures. I doubt other brain regions may also be affected in these pups. Did they perform a systematic examination across the entire brain? Some previous reports suggested that deficits from the striatum and cerebellum may contribute to ASD. How about these regions?

Response 1: We would like to express our sincere gratitude for the reviewer's valuable feedback on our study. Your encouraging and thoughtful remarks have been immensely helpful in refining our work. We fully agree with your perceptive comments, and we can confirm that we have indeed measured the size of the striatum at E17 and P0 on brain slices. Our finding shows a decrease in the size of striatum in PE-exposed offspring at postnatal day 0. To support these results, we added the statistical data in Fig S2. C. Once again, we thank you for your constructive feedback and for taking the time to review our manuscript. Your input has been invaluable in enhancing the quality of our research.

Question 2: Still in Fig. 1: Is the decrease in cortical length due to the reduction in total cell numbers, or just only loss in the proliferating progenitors? How about the miss-alignment of newborn neurons (affecting neuronal migration)? The author should be able to show the analysis, or at least consider the above possibilities in the context.

Response 2: Thanks for the reviewer's suggestion. Our previous study on rats has demonstrated that prenatal exposure to preeclampsia does not significantly alter the laminar structure of the brain (PMID: 25575681). To assess the migratory ability of newborn neurons, we utilized the 24-hour BrdU labeling method, and our preliminary findings indicate that the migratory capacity of BrdU-labeled newborn neurons in cortex remains unaffected. In future, we aim to expand our research in this area and explore the potential impact of prenatal preeclampsia exposure on the migration and

connectivity of brain neurons in depth.

Question 3: Fig. 2: the authors identified a proliferation reduction in the PE-progenitors both in vivo and in vitro. What specific cell type is that? Radial glia, Intermediate progenitor cells (IPCs), neuronal precursors, or glia precursors? Is there any reduction in the total neuron number or glia?

Response 3: We greatly appreciate the reviewer's valuable suggestion. We would like to clarify that we utilized immunofluorescence analysis to identify the Nestin positive cells in most of the cells dissociated from the neurosphere. To support our findings, we have included a representative image and statistical results in Fig S2. D and E. As Nestin is a marker for radial glia, we postulated that the decreased proliferation of radial glia, instead of intermediate progenitor cells (Tbr2⁺ cells), may be responsible for the smaller cortex size observed in offspring exposed to PE. However, we could not distinguish between neuronal and glial precursors, since both cell types can originate from radial glia.

Our previous PE model in rats demonstrated a decrease in NeuN⁺ cells (a neuron marker) and an increase in GFAP⁺ cells (an astrocyte marker) in the adult rat brain (PMID: 25575681). Although the radial progenitors undergo a shift from a neurogenic to a gliogenic mode during late embryogenesis (approximately from E18.5 to P0 (PMID: 22998872), it was reported that astrocytes in the postnatal cortex in mice originate from the local proliferation of differentiated glia (PMID: 22456708). We speculate that the total number of neurons in the PE group may be reduced due to the inhibition of neurogenesis in the fetal brain in utero, while the total amount of glia in offspring from PE mothers may be elevated after delivery due to other unknown mechanisms caused by PE exposure. The imbalance between the number of neurons and glia could contribute to ASD and is an intriguing question to explore in future studies. We thank the reviewer once again for insightful comment, which inspired us to delve deeper into this area of research.

Question 4: I am not sure whether the authors examine adult PE-exposed mice brains. Is the PE brain normal? Size? Weight?

Response 3: Thanks for the reviewer's question. We would like to clarify that the weight of the brain in PE offspring is normal. We have added representative images and the corresponding statistical results in Fig S2. F and G.

Question 5: Fig 3: since the hippocampus region was also affected in PE mice, is there any deficit in hippocampus-related behavior? Learning disability?

Response 5: Yes, the reviewer is absolutely right. The hippocampus region also has been affected in PE mice. We have examined the spatial learning and memory using water maze test, which is associated with hippocampus region. Our results indicate that spatial learning and memory are significantly impaired in male offspring exposed to PE, compared to control male offspring. The data is presented in Fig S4. D and E, and we have added the relevant methods and figure legend to the Supplementary Materials and Methods section.

Question 6: Fig. 5 and Fig. S5: the author should state which cortical layer of neurons was analyzed in this research.

Response 6: Thank you for pointing out this omission. The pyramidal neurons from the layer II/III of the primary somatosensory cortex (S1) were captured and further analyzed. We have added this information in both the Methods section and the figure legend (Revised Fig S6).

Reviewer #2 (Comments to the Authors (Required)):

Thank you for the opportunity to review this interesting paper, which presents findings related to the role of an animal model of preeclampsia (chronic L-NAME) on in utero offspring brain programming relevant to ASD. This work constitutes a novel contribution to the field and will advance our knowledge of inflammatory mechanisms of neurodevelopmental programming by gestational diseases and exposures which impact placental health and function.

Question 1: Despite enthusiasm, some details are absent from the report and further rationale is required. Methods should be fully explained throughout, and some additional analyses considered, as described below. Additionally, please consider that some assessments (embryonic) might consider the biological replicate to be the litter rather than the individual animal. Language should also be revised throughout for clarity.

The data are supportive of the conclusion that maternal L-NAME leads to ASD-like phenotypes in offspring, and that some but not all of these are dependent on TNF α signaling.

Response 1: We would like to express our sincere appreciation for the encouraging and thoughtful remarks provided by the reviewer. After careful consideration of the reviewer's feedback, we have made substantial revisions to our manuscript. Specifically, we have re-assessed the embryonic data based on the litter and added more details to the Methods section. We hope that these revisions have strengthened the overall quality of the manuscript.

Some specific comments and suggestions are listed:

Abstract:

Question 2: The authors state "animal model and underlying mechanisms were not reported yet, hindering the development of effective therapeutic approaches." However, multiple studies have focused on the developing brain impacts of preeclampsia

Response 2: We highly appreciate the suggestion provided by the reviewer. Accordingly, we have made the necessary correction to the sentence as follows: 'However, the exact mechanisms underlying the impact of PE on progeny ASD are not fully understood, which hinders the development of effective therapeutic approaches'.

Question 3: "characteristics of ASD" should be revised to "ASD-like phenotypes" or "ASD-like characteristics"

Response 3: Following the reviewer's suggestion, we have made a revision in the Abstract to use the phrase 'ASD-like phenotypes'.

Introduction:

Question 4: Prevalence data for ASD should be updated to more recent figures (e.g.,

1:44 in the US, CDC), as rates change yearly and have risen since 2014.

Response 4: Thanks for the reviewer's comment. We have revised the sentence to read: 'The prevalence of ASD is increasing each year, with rates reaching approximately 1 in 44 children in the US.' We have also added a relative reference to support this statement.

Question 5: Line 36: The 17-50% contribution by environmental drivers of ASD should be cited.

Response 5: Your suggestion is greatly appreciated. We have added an original reference to support the statement.

Question 6: Lines 38, 48: You may also wish to cite reviews on the topic of PE impacts on neurodevelopmental disabilities and ASD (PMID: 35872512, PMID: 32209456)

Response 6: Thank you for the reviewer's recommendation. We have inserted the references PMID: 35872512 and PMID:32209456 in line 39 and 49, respectively.

Question 7: Line 57-58 requires a citation: " which results in preterm birth and escalates the risk for aberrant neurodevelopment in offspring."

Response 7: Thank you for reminding us. We have added a reference (PMID: 28689331) to support this sentence.

Question 8: Discussion of the existing literature linking animal models of PE to ASD-like phenotypes and neurodevelopmental deficits is lacking some references (e.g., line 64-65: lacks studies on PE animal models to mimicked key characteristics of ASD in offspring exposed to PE). Consider referencing: PMIDs 33510137, 30683649, 25575681. Many of these are summarized in PMID: 32209456.

Response 8: We appreciate the reviewer's thorough reading and helpful suggestions to improve our manuscript. In response to the reviewer's comments, we have not only added the recommended references but also revised the sentence to read: 'Although several PE animal models have been reported to demonstrate that offspring from PE mother indeed performed neurodevelopment and some behavior deficiency, it still lacks further studies on PE animal models to describe the ASD-like characteristics in offspring'.

Question 9: lines 67-68: Please specify PE-endophenotypes recapitulated by the chronic L-NAME model

Response 9: Thank you for pointing out this. We have specified PE phenotypes induced by L-NAME in Introduction. The sentence is revised to read: 'which could recapitulate almost aspects of preeclampsia pathogenesis, including sustained hypertension and proteinuria'. Additionally, we have added two references (PMIDs: 22615111 and 7909994) to support this statement.

Question 10: The rationale for focusing on TNF α should be more thoroughly described. For example, are other disorders of elevated TNF α in pregnancy linked to ASD?

Response 10: We appreciate the reviewer's suggestion and have incorporated a new paragraph to elaborate on the relationship between immune disturbances during pregnancy and the development of ASD in offspring, as well as the potential role of TNF α in the etiology of ASD. Although a strong correlation has been established between elevated TNF α expression and the occurrence of autism, the relationship between TNF α expression levels in pregnant mothers during pregnancy and the occurrence of autism in their offspring remains unclear. We believe that our work may be one of the first to comprehensively investigate the potential link between TNF α in pregnant mothers during pregnancy and the occurrence of autism in their offspring. The new paragraph is below: 'previous research has indicated that an imbalance in pro-inflammatory and anti-inflammatory cytokines in early pregnancy is a high-risk factor to the development of ASD in offspring. This altered cytokine profile in maternal circulation may affect fetal brain development through indirectly or directly pathways. It is believed that aberrant maternal immunity activation can disrupt normal fetal brain development processes such as neurogenesis and neuronal branching. Observation researches have shown several critical pro-inflammatory cytokines such as Interleukin-6 (IL-6), tumor necrosis factor- α (TNF α) are elevated in maternal immune activation model, which has been linked to ASD. In particular, the relationship between TNF α and autism is well-established in epidemiological research and animal models. However, no animal model has demonstrated whether common pregnancy complications such as preeclampsia or gestational diabetes increase the susceptibility of offspring to ASD through abnormal expression of TNF α .'

Methods:

Question 11: How old were dams and sires at breeding?

Response 11: Thanks for your inquiry. We use 8-week-old dams and sires when breeding, and we have included this information in the Methods section for clarity.

Question 12: Was water consumption altered in the L-NAME group?

Response 12: Thanks for your feedback. We would like to clarify that we did not observe any significant changes in water consumption in the L-NAME group.

Question 13: What concentration of IgG was administered?

Response 13: Thanks for your comment. To clarify, the dose of IgG used in our experiments is 1.5 μ g/g body weight, which is equivalent to the dose of TNF α antibody used in our experiments. This information has been included in the Methods section.

Question 14: There is some concern with the use of photoplethysmography with dark skin/hair (e.g., C57Bl/6 animals). This should be briefly mentioned/discussed.

Response 14: We appreciate for reviewer pointing out this potential limitation of

using photoplethysmography to measure blood flow changes in animals with darker skin or fur. At the beginning of our study, we encountered difficulties in measuring these changes, but we strictly followed the recommendations provided from the company, cleaned the tails with 100% ethanol, and used non-treated pregnancy C57Bl/6 mice and CD-1 mice as a reference for pilot experiments. Once the results become more reproducible, we got started performing our experiments. Combined with other phenotypes, such as proteinuria, placenta deficiency, in L-NAME treated mice, we believe these procedures could minimize variants and increase the accuracy of our data. It is worth noting that the BP-2000 blood pressure analysis system, which we used in our study, has been cited in more than 500 research papers and is recognized as the world's leading non-invasive blood pressure analyzer for mice and rats. We have discussed our methodology, including the limitations and references, in the Supplementary Materials and Methods.

Question 15: Handling and restraint stress related to photoplethysmography may also be considered a mild chronic stressor, which may interact with L-NAME in pregnancy. This should also be discussed.

Response 15: We acknowledge the reviewer's valid concern regarding the potential mild chronic stress caused by handling and restraint stress during our blood pressure measurements using photoplethysmography. While we could not completely eliminate this factor, we took several measures to minimize its impact. Firstly, we trained the animals for three days before the L-NAME treatment. Secondly, we performed the measurements at the same time each day. Thirdly, we handled the animals gently and did our best to keep them calm throughout the procedure. Finally, we ensured that the measurement sessions were conducted in a quiet room, free from any loud noises or distractions. Notably, we handled the mice in both the control and L-NAME groups identically. We have detailed these concerns in the Discussion section. The sentences are revised to read: 'Finally, it is important to acknowledge the limitations of our current study. One such limitation is the potential mild chronic stress resulting from the handling and restraint required when measuring blood pressure in mice. Although we took several steps to mitigate this effect - including a three-day training period before L-NAME treatment, consistent measurement times, gentle handling of the animals, and conducting measurements in a quiet room - it is possible that this stressor could still interact with L-NAME in pregnancy.'

Question 16: The breeding scheme is unclear. The authors write "Offspring from control and PE groups were bred with their mothers and weighted using an electric scale to trace their weight trajectories." Does this mean that the ordered animals were not used in experiments, but rather their progeny, who were exposed to L-NAME in utero and then when they themselves were pregnant? This would be an inter-generational model and different from what is described elsewhere in the paper.

Response 16: We apologize for the confusion caused by the previous sentence to the reviewer. The ordered dams and sires were used for breeding in our experiments. We have rephrased it to better convey our intended meaning in the Method section.

Specifically, we aimed to clarify that the offspring from both the control and PE groups were raised by their respective mothers and their weight trajectories were monitored using an electric scale.

Question 17: Were litters culled? This may be a source of variance in rearing experience of offspring. What was the average litter size in each group at behavioral testing?

Response 17: Thanks for the thoughtful suggestions. To clarify, no litters were culled in our study. While we highly respect the reviewer's opinion that rearing experience could cause variance, we partially agree with this assessment. Most research on autism indicates that mothers' parenting behavior is not a direct risk factor for the condition, which is likely caused by a combination of genetic and environmental factors. While a mother's behavior can influence a child's development, well-being, and improve the behavior in ASD children in various ways, blaming mothers for their child's autism is unsupported by scientific evidence (PMID: 30658339).

However, we do believe that maternal milk may significantly impact offspring during the postpartum period, as the L-NAME used in our study may have long-term effects on maternal health. Supporting this, a study (PMDI 24292233) showed that chemokine levels in milk, such as IP-10, MCP-1, and MCP3, were reduced in hematopoietic TNF α knockout maternal mice, resulting in increased postnatal hippocampal proliferation and improved adult spatial memory in offspring. Therefore, we hypothesize that differences in spatial memory between control and PE offspring in our study may be due to milk chemokines. To further investigate this, we plan to incorporate cross-nursing between control and PE groups in our future projects to exclude the potential effects of rearing experience and breast-feeding during the postnatal period on development of offspring.

The average litter size was 5 in control and PE group. In general, the litter size is applied in many literatures on autism studies. At behavior testing, the offspring were selected from 4 to 6 litters in each group.

Question 18: For the three-chamber social task, stranger mice should ideally be sex, age, and size-matched.

Response 18: We thank your valuable feedback. The sex, age and size matched wildtype mice were selected as stranger mice in the three-chamber social task. We have revised these descripts in Supplementary Materials and Methods.

Question 19: What software was used for each task and for behavioral quantification? If scoring was manual, was it repeated and tested for inter-rater reliability?

Response 19: Thank you for the reviewer's inquiry. With the exception of the marble burying assay, all behavioral tests were analyzed using the Noldus software, which automatically calculated the behavioral data. The number of buried marbles was manually counted according to the criteria described in the reference (PMID: 26822608) and it was tested for inter-rater reliability (two independent experimenters determined the results based on the criteria). We have provided a detailed description

of the methods used, along with the references, in the Supplementary Materials and Methods section. The revised methodology in the Supplementary Materials and Methods section is now in line with these details.

Question 20: What was the standard housing bedding material for these animals? Consider thoroughly reporting all rearing/housing conditions and manipulations per the guidelines outlined in PMID: 30188509. You may attach their helpful template table as a supplement to your paper.

Response 20: We greatly appreciate the reviewer's suggestion. To address their concern, we have included a supplemental material that describes in detail the housing conditions and experimental manipulations, which were conducted following established guidelines (PMID: 30188509).

Question 21: Supplement IF section: What is meant by "enough PBS and 4% PFA" ?

Response 21: We apologize for the confusing sentence. To achieve a good fixation result, we usually perfuse with 5 ml of PBS first, followed by 5 ml of 4% PFA via heart perfusion. We have now corrected the sentence in the Supplementary Materials and Methods-IF section.

Question 22: How many dams were bred per condition? Were resorptions detected (at E15, E17)? Why were E15.5 and E17.5 selected as timepoints for embryonic brain analyses? Some rationale would be helpful.

Response 22: Fifty dams for each condition were utilized in this project. The precise number of dams utilized in a particular experiment is provided in the figure legends. No significant resorptions were detected at E15 or E17. The selection of E15.5 and E17.5 as timepoints for embryonic brain analyses was likely based on their correspondence to specific stages of neurodevelopment in mice. At E15.5, the mouse brain is undergoing a rapid period of neuronal proliferation and migration, and the major brain structures have already been established. This timepoint is often used to examine changes in neuronal differentiation and migration, as well as the formation of synaptic connections. At E17.5, the mouse brain is also undergoing a period of significant growth and maturation, with continued neuronal migration and differentiation, as well as the formation of more complex neuronal circuits. This timepoint is often used to examine changes in neurogenesis, gliogenesis, and the maturation of neuronal circuits.

Importantly, some studies have found that individuals with autism may have altered patterns of neurogenesis, particularly in the prefrontal cortex and hippocampus (PMID: 27014681). Additionally, studies have suggested that there may be disruptions in the formation and function of neuronal circuits in individuals with autism, particularly in the areas of social cognition and communication (PMID: 28729065). In addition, previous studies have explored the effects of preeclampsia on postnatal day 0 neurodevelopment in a rat PE model (PMID: 25575681).

The gestational period of C57 mice is approximately 19.5 days, and the timepoint of drug administration is at gestational day 12.5. Therefore, we believe that gestational

day 15 and 17 are good timepoints to investigate the effects of preeclampsia on fetal neurodevelopment in a mouse model. This rationale has been added to the manuscript from line 126 to 129.

Question 23: Were cytokine levels normalized to total protein in serum?

Response 23: Thank you for the reviewer's comment. As stated in the manual of Mouse TH17 array1 (Raybiotech, QAM-TH17-1, GA, USA), specific cytokine standards in each array were predetermined to generate a standard curve for each cytokine. In our experiment, we added standard cytokines and samples to each array, followed by a sandwich ELISA procedure simultaneously. The cytokine concentration in the samples was determined by comparing the signals to the standard curve. Additionally, the volume of each serum sample is identical. Moreover, given the high levels of serum albumin and immunoglobulin in serum, it is not necessary to measure the total protein amount in serum.

Results:

Question 24: Average litter-wise brain weight and cortical thickness data should be provided (with averages per litter within each condition), as the litter may be considered the biological replicate.

Response 24: Thanks for your valuable suggestions. We have incorporated the necessary corrections to the data related to brain weight and cortical thickness by performing a litter-wise analysis. We have also updated the corresponding figure legend (line 861, 960 and 977) and statistical results (figure 1.A and C, fig S1.E, fig S2. B and C, fig S3. E-G) to reflect these changes.

Question 25: Line 83: "12 days after pregnancy" suggests a single post-partum manipulation. This should be revised to "at gestational day (GD) 12 through..." or similar.

Response 25: Following the reviewer's suggestion, we have revised it to "on gestation day (GD) 12 through" in line 106.

Question 26: Consider breaking up the results section into further subsections for improved clarity and flow

Response 26: The reviewer's suggestion is highly appreciated. We have revised the Results section by breaking it up into further subsections, which we hope will improve the clarity and flow of the manuscript.

Question 27: Lines 185-187: If these samples were paired, were placenta Tnfaip3 and brain NFkB correlated across PE or control (or all) fetuses?

Response 27: Thank you for bringing this up. Paired studies may be crucial for this type of analysis. However, at the moment, our samples were not paired. Because the results of the placenta here mainly serve to illustrate that the fetal tissues have a significant response to TNF α in maternal serum. We do not emphasize that TNF α in

the mother's serum affects fetal brain development through the placenta. Therefore, the current data does not require a matched study and does not affect our conclusion. Meanwhile, we speculate that TNF α may affect fetal development indirectly through first affecting placental function, as well as directly passing through the placenta.

Question 28: Line 189: E17.5 cortical array data should be shown in possible. Were these below limit of detection, or were they truly unchanged?

Response 28: We appreciate the suggestion made by the reviewer. The cytokines of cortexes could be detected using the array. we have included E17.5 cortical array data in Figure 4A. The fetal cortex did not show any significant changes. We have updated the figure legend, methods and the statistical results sections.

Question 29: Was NPC soma size/volume changed by NFkB manipulations?

Response 29: Thanks for the reviewer's inquiry. We captured the images of NPC from control, L-NAME, TNF α and TNF α +Bay, and then measured the soma size using image J software. We did not find significantly changes in soma size. The data are performed in Fig S7. We have also described the methodology in Materials and Methods section.

Question 30: Line 215: "stored" should be changed to "restored"

Response 30: We apologize for the mistake. And we have fixed it. Thank you for bringing it to our attention.

Question 31: EPM: Open/Closed arm time should be calculated and reported

Response 31: Thank you for pointing out it. We have added the data in Fig S4. A and S8. A. Associated figure legend has been added.

Question 32: Social: A sociability quotient (social-nonsocial/total) might be also be calculated and reported to normalize social time to time spent with the empty cup.

Response 32: The reviewer's suggest is absolutely right. We have calculated a sociability quotient (social-nonsocial/total time in middle chamber) and preformed the data in Fig S4.C and S8.B. The results and conclusions remain unchanged. Relative figure legend and methodology have been added.

Question 33: Open field: total distance traveled should be shown in figures.

Response 33: The reviewer's suggestion is highly appreciated. We have exhibited the data on total distance in open filed test in Fig S4. B and S8. C. We also added the corresponding figure legend, methodology and statistic results.

Figures:

Question 34: Fig 2C: "Percentage" is misspelled. This should be specified as "percent of X"

Response 34: We apologized the misspelling. We have revised it to "Cumulative percent of neurosphere diameters" in both Fig 2C and relative figure legend.

Question 35: Fig 2E: down and up regulated in reference to what? This should be specified in the legend

Response 35: We are sorry for the omission. The down and up regulated differential expressed genes (DEGs) in reference to control samples. We have fixed it in the legend.

Discussion:

Question 36: The TNF α antibody rescue experiment occurs at PD16.5, which is well after placentation processes have occurred. Placentation abnormalities are often credited with driving many of the physiologic disruptions in preeclampsia. How might the authors explain this limitation or caveat in their interpretation?

Response 36: Thank you for pointing this out in the review. The placenta not only provides the interface for gas and nutrient exchange, as well as immunological barrier between the mother and the fetus, but also secretes hormones to support pregnancy. We greatly appreciate the recognition of abnormal placental development's role in preeclampsia onset. The literature indicates that placental developmental abnormalities, such as inadequate invasion of trophoblast cells, are significant causes of preeclampsia, and the delivery of the placenta is the most effective treatment for PE. We understand that placental development is generally completed by PD12.5 in mice (PMID: 24286824). Previous studies indicate that L-NAME treatment in mice can alter placental morphology after PD12.5 (PMID: 35152352 and PMID: 18246055). Therefore, starting TNF α antibody treatment on PD16.5 may prevent the restoration of placental function, which could explain why we were unable to improve intrauterine growth restriction in mice. However, we observed that anti-TNF α was able to rescue ASD behaviors in PE-exposed offspring. We have added this discussion from line 296 to 306.

Question 37: Line 254: "Deeper mechanisms should be figured out in future" is a non-specific statement. What might some of these mechanisms for future consideration be? Several should be suggested.

Response 37: Thank you for the reviewer's comment. We have added potential molecular mechanisms to the discussion section, from line 289 to line 294. For instance, it would be valuable to investigate the effects of different phosphorylation sites of NF κ B (such as Ser 536 or S468) on neural progenitor cell proliferation or neurite branching. Alternatively, TNF α -induced changes in NF κ B signaling could potentially result in alterations in the epigenetic landscape of neural progenitor cells and synaptic development, such as DNA methylation and histone modifications.

Question 38: The clinical literature on use of anti-TNF α agents should be briefly discussed, particularly as it pertains to impacts on gestation and offspring brain outcomes (e.g., PMID: 34489011)

Response 38: Thanks for the reviewer's valuable suggestion. We have incorporated it in the manuscript in line 306 to line 317. The discussion paragraph is below "And we

believe that anti-TNF α treatment would be beneficial for both the mother and fetuses. This is because a number of anti-TNF agents, including infliximab (IFX), etanercept (ETA), adalimumab (ADA), certolizumab pegol (CZP), and golimumab (GOL), are widely used to treat inflammatory diseases such as inflammatory bowel disease (IBD), rheumatoid arthritis (RA), ankylosing spondylitis (AS), psoriatic arthritis (PsA), and psoriasis (PMID: 23964937). Most studies have reported that the use of anti-TNF agents during pregnancy in patients with IBD, RA, AS, PsA, and psoriasis does not increase the occurrence of adverse maternal or infant outcomes (PMID: 34489011), although there is a higher risk of infections in patients exposed to anti-TNF α (PMID: 35687009). The risk and benefit balance for both the PE mothers and offspring's health requires further investigation and discussion in future studies.”. Thank you again for your valuable comments.

Question 39: Do you predict that your results pertain to a subset of ASD patients, for whom prenatal inflammation is a driving pathoetiologic factor? ASD is a highly variable, spectral disorder with many different potential causes/influences. Does your work highlight a need for improved individualized treatments/preventions in ASD?

Response 39: Thank you for your valuable inquiry. Yes, we believe that our results pertain to a subset of ASD patients. Because our research and others' results strongly suggest that maternal pro-inflammation increases the risk of ASD in offspring, we cannot conclude that this applies to all cases of ASD, as it is a highly complex and variable disorder with many different potential causes and influences. It is possible that our findings only pertain to a subset of ASD patients who were exposed to prenatal inflammation. Therefore, further studies are needed to confirm the relationship between prenatal inflammation and ASD in humans.

As for the need for improved individualized treatments and preventions in ASD, our work emphasizes the importance of identifying and addressing specific risk factors that contribute to ASD. If prenatal inflammation is identified as a key risk factor for ASD in certain individuals, then individualized prevention and treatment strategies could be developed to target this factor. In light of the complex nature of ASD, individualized approaches to understanding and treating the disorder are likely to be the most effective. Thank you again for your insightful inquiry.

May 23, 2023

RE: Life Science Alliance Manuscript #LSA-2023-01957R

Dr. Wenlong Zhao
Rutgers, The State University of New Jersey
170 Frelinghuysen Rd
Piscataway 08854

Dear Dr. Zhao,

Thank you for submitting your revised manuscript entitled "Preeclampsia promotes autism in offspring via maternal inflammation and fetal NFκB signaling". We would be happy to publish your paper in Life Science Alliance pending final revisions necessary to meet our formatting guidelines.

- please address the remaining Reviewer 2's concerns
- please update the manuscript to include the author listing and affiliations on the first page
- please add an Author Contributions section to your main manuscript text
- please incorporate any points from the Conclusion section into the Discussion, we only allow a Discussion section
- please add ORCID ID for secondary corresponding author--you should have received instructions on how to do so

Figure Check:

- please make sure each Table has a title

A. FINAL FILES:

B. MANUSCRIPT ORGANIZATION AND FORMATTING:

Sincerely,

Reviewer #1 (Comments to the Authors (Required)):

It's very nice to see most of my questions were addressed properly in the revised version. I don't have any other concerns. I would like to suggest the editor publish this manuscript soon.

Reviewer #2 (Comments to the Authors (Required)):

Thank you for providing revisions to your interesting manuscript. A few concerns remain:

- While the addition of a paragraph on MIA and ASD in the introduction is welcome, it remains unfocused on TNF α and ASD. Please elaborate further on the following, providing more specifics: "In particular, the relationship between TNF α and autism is well-established in epidemiological research and animal models"
- water consumption data should be added
- IUGR stands for Intrauterine growth restriction, not retardation
- You may wish to emphasize no significant difference in litter size between groups
- Please note that many ELISA-based approaches to protein measurement do normalize for total protein. It is a significant limitation that only absolute levels of cytokine (and not relative to total protein) are included. If possible, you may wish to at the very least demonstrate that total serum protein did not differ by condition.
- correction for litter in embryonic measures (weight, thickness) remains unclear- I originally suggested that you may wish to run your tests on a litter-wise basis (e.g., collapse all individual measures to their litter-wise mean and then compare). However, if you are underpowered to do that given only 4 litters per time-point it's alright to exclude. It is helpful that you have included the average pups per litter, however.

Reply to Reviewers

Please see Reviewers' comments below in **black** and also our reply **marked in yellow**.

Reviewer #1 (Comments to the Authors (Required)):

Question 1: It's very nice to see most of my questions were addressed properly in the revised version. I don't have any other concerns. I would like to suggest the editor publish this manuscript soon.

Response 1: Thank you for your positive feedback on the revised version of our manuscript. We are glad to hear that most of your questions were addressed properly. Your suggestion to publish the manuscript is greatly appreciated.

Reviewer #2 (Comments to the Authors (Required)):

Thank you for providing revisions to your interesting manuscript. A few concerns remain:

Question 1: While the addition of a paragraph on MIA and ASD in the introduction is welcome, it remains unfocused on TNF α and ASD. Please elaborate further on the following, providing more specifics: "In particular, the relationship between TNF α and autism is well-established in epidemiological research and animal models"

Response 1: We appreciate the reviewer's valid concern regarding the association between TNF α and autism. Upon careful consideration, we recognize that the existing literature does not provide conclusive evidence for a strong association between TNF α and autism. Therefore, we have rephrased the statement to avoid overstatement and potential confusion. Furthermore, we have incorporated specific examples that highlight the relationship between TNF α and autism. We believe that these revisions in the Introduction section, along with the corresponding references, provide a more accurate representation of the relationship between TNF α and autism: 'Several studies, including both epidemiological research and animal models, have suggested an association between TNF α and autism. For instance, Xiang Yu et al. conducted a study where they measured plasma levels of cytokines in children with autism spectrum disorder (ASD) and typically developing children. They found elevated levels of TNF α in male children with ASD, which positively correlated with their total development quotient. Another study by Jones et al. demonstrated that higher levels of mid-gestational cytokines in maternal serums, including TNF α , were associated with an increased risk of ASD with intellectual disability compared to developmental delay without ASD (Jones KL et al, 2017). Moreover, elevated levels of TNF α have been observed in a mouse model of autism induced by prenatal exposure to valproic acid, suggesting a potential contribution of TNF α to autism in mice'.

Question 2: water consumption data should be added

Response 2: Thank you for reviewer's suggestion. We have added the water consumption data in new Fig S1. D and Fig S11. I, along with the corresponding figure legend and methodology in main manuscript and supplementary materials and methods.

Question 3: IUGR stands for Intrauterine growth restriction, not retardation

Response 3: Thank you for pointing out it. It has been fixed.

Question 4: You may wish to emphasize no significant difference in litter size between groups.

Response 4: The reviewer's suggest is absolutely right. We have now emphasized in line 123 of the main manuscript that there is no significant difference in litter size between the groups.

Question 5: Please note that many ELISA-based approaches to protein measurement do normalize for total protein. It is a significant limitation that only absolute levels of cytokine (and not relative to total protein) are included. If possible, you may wish to at the very least demonstrate that total serum protein did not differ by condition.

Response 5: The reviewer's suggestion is highly appreciated. We have determined the concentrations of total serum protein using the Biuret Method. The data has been added to Fig. 4A. The corresponding figure legend and methodology in the main manuscript and supplementary materials and methods have also been included. The concentrations of cytokines in the serums were normalized to their respective total protein concentrations. The reanalyzed data does not alter the results and conclusions.

Question 6: correction for litter in embryonic measures (weight, thickness) remains unclear- I originally suggested that you may wish to run your tests on a litter-wise basis (e.g., collapse all individual measures to their litter-wise mean and then compare). However, if you are underpowered to do that given only 4 litters per time-point it's alright to exclude. It is helpful that you have included the average pups per litter, however.

Response6: Thank you for your valuable feedback and suggestions regarding the correction for litter in the embryonic measures of weight and thickness. We greatly appreciate your recommendations and have carefully considered them. In response to your suggestion, we have analyzed the data on a litter-wise basis by averaging individual measures within each litter and comparing them. For example, in the weight measurement experiment, we analyzed data from 8 litters. We acknowledge that the sample size of 4 litters per time-point may have limited power for certain specific analyses. However, it is important to note that we collected 2 pups as replicates from each litter, resulting in a total of 8 pups per group. Additionally, our study encompassed 3 developmental stages and involved the analysis of cortical thickness at different cross-sections, as well as the examination of proliferation-related markers. Taking all of these factors into account, we believe that our methodology provides a representative and comprehensive understanding of the impact of preeclampsia on offspring brain development. We sincerely appreciate your recognition and support, as they hold great value for us and contribute to our ongoing growth and advancement in research. We wholeheartedly thank you for your invaluable contribution and support!

May 30, 2023

RE: Life Science Alliance Manuscript #LSA-2023-01957RR

Dr. Wenlong Zhao
Rutgers, The State University of New Jersey
170 Frelinghuysen Rd
Piscataway 08854

Dear Dr. Zhao,

Thank you for submitting your Research Article entitled "Preeclampsia promotes autism in offspring via maternal inflammation and fetal NFκB signaling". It is a pleasure to let you know that your manuscript is now accepted for publication in Life Science Alliance. Congratulations on this interesting work.

DISTRIBUTION OF MATERIALS:

Again, congratulations on a very nice paper. I hope you found the review process to be constructive and are pleased with how the manuscript was handled editorially. We look forward to future exciting submissions from your lab.

Sincerely,
